# Novel Cell Receptor System of Eukaryotes Formed by Previously Unknown Nucleic Acid-Based Receptors

**Victor Tetz** [1] and **George Tetz** [1,*]

Department of Systems Biology, Human Microbiology Institute, New York, NY 10128, USA
* Correspondence: g.tetz@hmi-us.com

**Abstract:** Here, our data provide the first evidence for the existence of a previously unknown receptive system formed by novel DNA- and RNA-based receptors in eukaryotes. This system, named the TR-system, is capable of recognizing and generating a response to different environmental factors and has been shown to orchestrate major vital functions of fungi, mammalian cells, and plants. Recently, we discovered the existence of a similar regulatory system in prokaryotes. These DNA- and RNA-based receptors are localized outside of the membrane forming a type of a network around cells that responds to a variety of chemical, biological, and physical factors and enabled the TR-system to regulate major aspects of eukaryotic cell life as follows: growth, including reproduction and development of multicellular structures; sensitivity to temperature, geomagnetic field, UV, light, and hormones; interaction with viruses; gene expression, recognition and utilization of nutrients. The TR-system was also implicated in cell-memory formation and was determined to be responsible for its maintenance and the forgetting of preceding events. This system is the most distant receptive and regulatory system of the cell that regulates interactions with the outer environment and governs the functions of other receptor-mediated signaling pathways.

**Keywords:** receptors; Teazeled receptor; TezR; cell surface-bound nucleic acids; cell surface-bound DNA; cell surface-bound RNA; extracellular nucleic acids





## 1. Introduction

Eukaryotes are a highly diversified group of unicellular and multicellular organisms represented by protozoans, algae, fungi, plants, and animals, and their survival depends on their responses to an array of physical, chemical, and biological stimuli. Cell reception enables eukaryotic cells to sustain themselves through various environmental cues and adapt to changes of external signaling [1–3]. Moreover, it is a key element of cell-to-cell communication across multiple cell types and tissues, as well as inter-kingdom communication [4–7].

The known surface eukaryotic cell receptors are formed by proteins that can be grouped into three major classes, ion channel-linked receptors, G protein-coupled receptors (GPCR), and enzyme-linked receptors [8–10]. The GPCR are broadly distributed across nearly all tissues of macro-organisms [11]. They mediate cellular responses to metabolites, cytokines, hormones, and neurotransmitters, and thus act as key regulatory elements in a broad range of normal and pathological processes [12,13]. Receptors perceive environmental signals and transmit this ligand-binding event into signals for intracellular targets, causing a wide array of cellular responses, including the regulation of gene expression, cellular development and specification, tissue differentiation, cell migration, and many others [3,14–16]. Furthermore, abnormal receptor activation in humans is strongly associated with developmental abnormalities, aging, and cancer [15,17–19].

However, the particular aspects of molecular mechanisms that enable non-specialized eukaryotic cells to sense and respond to many external physical and chemical factors remain unclear [20]. For example, the ability of cells to respond to the Earth's magnetic

field and the formation of a circadian rhythm are believed to be attributed to the presence of magnetic field-sensitive molecules such as cryptochromes [21]. These molecules are found only in retinal cells of some animals; however, existing studies have not explained why even cells without these elements are known to sense and react to altered geomagnetic fields [22–25]. Some authors suggest that intracellular DNA molecules can interact with magnetic fields, but the underlying mechanisms of such sensing and interactions remain unclear [26–28].

The mechanism and regulation of photoreception in non-specialized cells also leave multiple unknowns. The most well-known light-sensitive molecules, opsins and cryptochromes, are not found in many cells that still can detect and respond to light [29–32]. The same is true for temperature sensing, since the tuning of cell growth dependent on the temperature and even the initial steps underlying the triggering of heat shock remain insufficiently studied [33]. Temperature-sensing structures are attributed to intracellular RNA molecules; however, they are believed to mediate responses to temperature changes and not to provide direct temperature-sensing functions [34–36].

Therefore, the particular receptors that enable eukaryotic cells to sense and respond to many factors, including xenobiotics, have not yet been fully characterized [37–39].

The reception and sensing of eukaryotes is tightly linked to cell memory, which also includes immunological memory [40–42]. Cell memory that can be short- and long-term is a part of a history-dependent ability of the cells to respond more rapidly and effectively to repeated stimuli [43,44]. It is obvious that cell reception plays a keen role in cell memory-formation; however, the molecular processes that drive the formation of cell memory remain elusive [45–47].

Taking a broad range of stimuli whose reception cannot be well explained with known receptors we studied the receptive and regulatory roles of extracellular DNA and RNA, since the novel roles, of nucleic acids that go beyond the processing of genetic information and protein synthesis, including the interaction with proteins, were recently described [48–50]. We discovered that prokaryotes have a previously unknown receptive system that is responsible for the interaction of cells with the environment [51]. Here, we show the existence of a similar, previously unknown TR-receptive system in eukaryotes that manages interactions between cells and the environment. As in prokaryotic cells, this system was determined to consist of novel elements formed by DNA or RNA molecules, located outside the cell membrane, with receptive and regulatory functions. Owing to the similarity with spiny bracts of *Dipsacus* spp., we named them Teazeled (common name of *Dipsacus* spp.) receptors or "TezRs". We propose that these TezRs together with reverse transcriptases and integrases form a "Teazeled receptor system" or TR-system. In this work, we describe an array of environmental stimuli being regulated by the TR-system and show its implication in cell memory formation and forgetting.

## 2. Materials and Methods

### 2.1. Cell Lines, Culture, and Treatments

The cell lines VERO CCL-81, HEK293, CHO-K1, and T98G (American Type Culture Collection, ATCC) were used in this study. All cell lines were grown in a humidified chamber (95% air, 5% $CO_2$) at 37 °C and were authenticated using the short tandem repeat profiling method at our institute. Cells were cultured in Dulbecco's modified Eagle's medium (DMEM; Sigma-Aldrich, St. Louis, MO, USA), supplemented with 10% fetal bovine serum (FBS; Thermo Fisher Scientific, Waltham, MA, USA),1 mM L-glutamine (Sigma-Aldrich, St. Louis, MO, USA), penicillin G (100 U/mL), and streptomycin (100 mg/mL; Sigma-Aldrich, St. Louis, MO, USA), hereafter referred to as complete DMEM. The medium was changed every 2–3 days, and cells were subcultured when they reached 80–90% confluence. Cells at passages 6–10 were used for the experiments. Trypsin/EDTA (0.05%, Invitrogen) was used to detach the cells from the flask, either for passaging or for any experiment. The wild-type HSV-1 strain KOS was propagated in Vero cells.

Candida albicans VT 298 was obtained from the collection of Human Microbiology Institute (NY, USA). Fungi were passaged weekly on Sabouraud agar (Oxoid, UK) and stored at 4 °C. All subsequent liquid subcultures were derived from colonies isolated from these plates and were grown in Sabouraud broth (Oxoid, UK) at 37 °C for 24 h if not stated otherwise. The other liquid medium used was M9 minimal salts (Sigma-Aldrich, MO, USA). For the experiments on solid media, fungi were cultured on Sabouraud agar (Oxoid, UK). All cultures were incubated in a Heracell 150i (Thermo Scientific, Waltham, MA, USA) or Sanyo MCO-19AIC (Sanyo, Japan).

### 2.2. Reagents

Bovine pancreatic DNase I with a specific activity of 2200 Kunitz units/mg and RNase A (both from Thermo Fisher Scientific, Waltham, MA, USA), or human recombinant DNase I (Catalent, Madison, WI, USA) were used at concentrations ranging from 1 to 50 µg/mL. Etravirine, tenofovir, lamivudine, raltegravir, and maltose were obtained from Sigma-Aldrich (St. Louis, MO, USA) and insulin-transferrin-selenium was from Becton Dickinson (cat. no. 354352; Franklin Lakes, NJ, USA).

### 2.3. Destruction of Primary TezRs in Planktonic Cultures

For the destruction of primary TezRs on fungal or mammalian cells, cells were harvested by centrifugation at $4000 \times g$ for 15 min (Microfuge 20R, Beckman Coulter), and the pellet was washed twice in PBS (pH 7.2; Sigma-Aldrich) or nutrient medium to obtain an optical density from 0.003 to 0.5 at 600 nm. Cells were treated with nucleases at 10 µg/mL at 37 °C for 10 min, unless stated otherwise. After treatment, nucleases were washed out three times in PBS or broth, and cells were centrifuged at $4000 \times g$ for 15 min (Microfuge 20R, Beckman Coulter), followed by resuspension in PBS or broth. Control cells were processed in the same manner; however, instead of treatment with nucleases, they were treated with water (used as a vehicle for nucleases).

### 2.4. Destruction of Primary TezRs of Mammalian Cells within Monolayers

For the destruction of primary TezRs on cell monolayers, DMEM was removed, and cells were treated with nucleases at 10 µg/mL at 37 °C for 10 min to generate TezR–D1$^d$, TezR–R1$^d$, or TezR–D1$^d$/R1$^d$ cells in DMEM without FBS. Then, cell monolayers were washed three times with PBS to eliminate remaining nucleases.

### 2.5. Destruction of Secondary TezRs

For the study of secondary TezRs, *C. albicans* was grown on 1.5% Sabouraud agar supplemented with nucleases. Sabouraud agar, after autoclaving at 121 °C for 20 min, was cooled to 45 °C and DNase I, RNase, or both was added at 10 µg/mL and mixed; then, 20 mL of the solution was poured into each 90 mm glass Petri dish. Control *C. albicans* were cultivated on the agar without nucleases.

### 2.6. Inactivation of TezRs with Propidium Iodine

To inactivate primary TezRs with propidium iodine (PI), Vero cells were harvested by centrifugation at $4000 \times g$ for 15 min (Microfuge 20R; Beckman Coulter, La Brea, CA, USA). The pellet was washed twice with PBS at pH 7.2 (Sigma-Aldrich). Cells were treated with PI (final concentration: 1 µM) for 30 min at 37 °C. The PI-treated cells were washed three times in PBS, centrifuged at $4000 \times g$ for 15 min, and resuspended in PBS or nutrient medium. Controls were similarly treated with water (used as a vehicle for nucleases).

### 2.7. Morphology Index

The mean morphology index (MI) was used to evaluate *C. albicans* morphology (Merson-Davies and Odds, 1989). Briefly, MI represented the following morphologies: MI 1 (0–1.5), yeast cells, single or small groups; MI 2 (1.5–2.5), elongated, ovoid cells ('short pseudohyphae'); MI 3 (2.5–3.4), elongated cells, pseudohyphal, obvious constrictions, sides

almost parallel; MI 4 (>3.4), true hyphal formation, parallel-sided, minimal constrictions at sites of septum formation. A minimum of 100 cells were observed for each morphological evaluation [52].

### 2.8. Growth Curve

For growth rate determination at all measured time points, planktonic growing stationary-phase *C. albicans* were centrifuged $4000 \times g$ for 15 min (Microfuge 20R; Beckman Coulter, La Brea, CA, USA). The pellet was washed twice with PBS at pH 7.2 (Sigma-Aldrich) to eliminate the extracellular matrix. PBS was added and cells were treated with nucleases at 10 µg/mL as described, and 5.5 log10 cells were inoculated into 4.0 mL of mixed Sabouraud broth (Sigma-Aldrich). Growth curves were generated by measuring the optical density at OD600 using a NanoDrop OneC spectrophotometer (Thermo Fisher Scientific, Waltham, MA, USA).

### 2.9. Fungal Viability Test

For all experiments used in this study for the evaluation of fungal viability, the *C. albicans* suspensions were serially diluted, and 100 µL of the diluted suspension was spread onto Sabouraud agar plates. Plates were incubated at 37 °C overnight, and CFUs were counted the next day.

### 2.10. Biofilm Morphology

For culture-based experiments on *C. albicans* biofilms, we prepared glass Petri dishes with Sabouraud agar supplemented or not supplemented with DNase I, RNase, or both at 10 µg/mL, as described previously. *C. albicans* was separated from the extracellular matrix by washing three times in PBS or broth and centrifuged each time at $4000 \times g$ for 15 min (Microfuge 20R, Beckman Coulter). Then, 25 µL 5.5 log10 cells were inoculated into the center of the agar and incubated at 37 °C for specific 7 days. Macroscopic biofilm images were taken with a digital camera, the Canon 6 (Canon, Japan), and analyzed with Fiji/ImageJ software [53,54].

### 2.11. Light Microscopy-Based Methods

Microscopic experiments were performed using a Zeiss Axiovert 40C microscope (Carl Zeiss, Oberkochen, Germany) with a ×10/0.25 objective microscope. Images were acquired using a Canon 6 (Canon, Japan). The cell area was analyzed for a total of 500 yeast cells using ImageJ/FIJI software and expressed in px2.

### 2.12. Fluorescence Microscopy

Fluorescence microscopy was used to confirm the destruction of primary TezRs with nucleases. For the propidium iodide (a membrane impermeant dye) assay, Vero cells after permeabilization were incubated with PI 1 µg/mL (Invitrogen, P3566) for 5 min on ice. At the end of staining, the cells were washed in triplicate with PBS. Cells were excited with 561 nm wavelengths and their emission was detected at 605 nm.

CHO cells were stained with cell-permeant 5 mM SYTO 9 dye that specifically stains the nucleic acids for 15 min in the dark. At the end of staining, the cells were washed in triplicate with PBS and studied with an excitation at 480 nm emission maximum at 503 nm. Cells were imaged using an EVOS FL Auto Imaging System (Thermo Scientific) equipped with a 60× or 100× objective and 2× digital zoom.

### 2.13. Flow Cytometry

Subconfluent cultures of Vero or Hek293 cells were collected, washed twice with DMEM without FBS, and resuspended in DMEM supplemented with FBS. DNase I and/or RNase were added at a final concentration of 100 µg/mL for 30 min to destroy the primary TezRs, as previously described. Cells were washed from nucleases and incubated for another 2.5 h in fresh DMEM with FBS at 37 °C, as previously described. Cells were

resuspended in PBS containing 0.2 μM YO-PRO-1 (Invitrogen, Y3603) and 1.5 μM PI (Invitrogen, P3566). In total, 10,000 cells were analyzed for each measurement.

The percentage of apoptotic cells was determined by flow cytometry using a CytoFLEX flow cytometer (Beckman Coulter, Brea, CA, USA). Cells undergoing apoptosis were stained with YO-PRO-1 but were impermeable to PI. Dead cells and cells in late apoptosis were permeable to both dyes. The results were expressed as the percentage of permeabilized cells. The experiment was performed in triplicate. Data were analyzed using FlowJo 10 software (Treestar Inc., Ashland, Wilmington, DE, USA).

### 2.14. Assays of RNase Internalization

The internalization of RNase was visualized in Vero cells and *C. albicans*. Vero cells were treated with fluorescein isothiocyanate (FITC)-labeled RNase A (50 μg/mL) at 37 °C for 30 min, as previously described [38]. Cells were washed three times with PBS to remove unbound proteins, and the fluorescence was measured either immediately or after 24 h of growth in FBS-supplemented DMEM (Axio Imager Z1, Carl Zeiss, Germany).

*C. albicans* cells were incubated with FITC-labeled RNase A (50 μg/mL) at 37 °C for 1 h, washed three times with PBS to remove unbound protein, and visualized either immediately or after 6 h of growth in Sabouraud broth (Axio Imager Z1, Carl Zeiss, Germany). To visualize the internalization of RNase into fungal biofilms, the biofilms of *C. albicans* were obtained as described previously and cultivated in media supplemented with fluorescein-labeled RNase, added at 50 μg/mL. After 24 h of growth at 37 °C, fungi were washed three times with PBS to remove unbound RNase and placed on a microscope slide, and fluorescence was monitored (Axio Imager Z1, Carl Zeiss, Germany).

### 2.15. Virus Release Assay

To analyze the effect of TezRs on HSV-1 release from cells, cell supernatant of 48 h-old Vero cells following HSV-1 infection (multiplicity of infection of 0.1) were collected. Cells were treated with DNase I, RNase or a combination of both 2 h post-infection to allow the virus to bind to the cell surface and internalize. The virus titer in the cell medium was determined by standard plaque assays using 10-fold serial dilutions of cell supernatants of Vero cells incubated for 48 h, after which cells were fixed and stained to count the plaques, as previously described [55].

### 2.16. Sensitivity of Cells to Opioids

Subconfluent cultures of T98G human glioblastoma cells, highly expressing opioid receptor, were collected, washed twice with DMEM without FBS, and resuspended in DMEM supplemented with FBS. Nucleases were added at a final concentration of 10 μg/mL for 15 min to destroy the primary TezRs, as previously described. The cells were seeded in 96-well plates at a density of 4.0 log10 cells per well and exposed to the freshly prepared tramadol (Sigma-Aldrich) at a concentration of 200 μM for 3 h at 37 °C with 5% $CO_2$.

### 2.17. Sensitivity of Cells to Insulin

CHO cells were initially serum-starved for 24 h and plated at a density of 4.2 log10 cells/well in 48-well culture plates. Cells were treated with the vehicle for nucleases to generate CHO control, CHO TezR–D1[d], TezR–R1[d], or TezR–D1[d]/R1[d] cells, as previously described, and treated with ITS-complex (insulin, 5 μg; transferrin, 5 μg; selenium, 5 ng/mL) according to the manufacturer's instructions (Sigma-Aldrich) in DMEM [56]. The number of attached cells was determined after 24 h of growth, according to previously established methods.

### 2.18. Wound-Healing Assay

After incubation in DMEM supplemented with 10% FBS for 48 h, Vero cells were starved in serum-free medium for another 24 h. Media were replaced with a fresh medium without FBS, and nucleases were added at 10 μg/mL for 15 min to generate TezR–D1[d], TezR–R1[d], or TezR–D1[d]/R1[d] cells. For some probes, PI, as previously discussed, was added.

The wounds were created using a pipette tip. The cells were then rinsed with medium to remove floating cells and debris. The culture plates were incubated at 37 °C in DMEM without FBS. The average width of the scratches at different intervals was determined, wounds were measured at 0, 6, 8, 12, and 24 h. Assays were repeated four times for each condition. Images were analyzed using ImageJ/Fiji software.

In some studies "relative wound closure" area was calculated (relative wound closure [%] = (Wound area at 0 h (pixel) − Wound area at certain hour (pixel)/Wound area at 0 h (pixel)) × 100% [57,58]. Experiments were repeated at least three times.

### 2.19. Modulation of Thermotolerance

Overnight *C. albicans* cultured in Sabouraud broth was washed with PBS to eliminate the extracellular matrix and diluted with PBS at an OD600 value of 0.5. Fungi were treated with nucleases, as previously described, to destroy primary TezRs. Control probes were treated with water used as a vehicle for nucleases, and cells were placed in 2 mL microcentrifuge tubes (Axygen Scientific Inc., Union City, CA) at 5.5 log10 CFU/mL. Each tube was heated to 37, 50, 51, 52, 53, 54, 55, 56, 57, 58, or 59 °C in a dry bath (Corning, LSE, Digital Dry Bath) for 15 min. After heating, some control *C. albicans* probes were immediately treated with nucleases to destroy primary TezRs or with water used as a vehicle for nucleases and washed three times to remove nucleases, as previously discussed. After that, all fungi were serially diluted and plated on Sabouraud agar, and the number of CFUs was determined at 24 h.

### 2.20. Modulation of Thermotolerance Restoration after TezR Destruction

To determine the time required for thermotolerance restoration in fungi following TezR destruction, overnight *C. albicans* cultures were treated with DNase I, RNase, or both at 10 µg/mL, as previously described. Next, fungi lacking TezRs were inoculated into Sabouraud broth and samples were taken hourly (each hour from 0 to 8 h) and heated at the maximum temperature that was tolerated by fungi lacking each TezR. Fungal viability was assessed as described in the "Modulation of thermotolerance" section. *C. albicans* treated with water (used as a vehicle for nucleases) was used as a control and was processed in the same way and heated at the lowest non-tolerable temperature. After that, fungi were serially diluted and plated on *C. albicans* agar, and the number of CFU was determined at 24 h. As the complete restoration of normal temperature tolerance, typical for control *C. albicans* (revealed as a disappearance of growth at higher temperatures) was observed, this experiment was not extended beyond this time point.

### 2.21. UV Assay

*C. albicans* was treated with nucleases to destroy the primary TezRs, as described. The control probes were treated with water. Fungi (9.5 log10 CFU/mL) were added to 9 cm Petri dishes with PBS and placed under a light holder equipped with a new UV light tube (TUV 30W/G30T8; Philips, Holland 254 nm) and irradiated for different time points at a distance of 50 cm. Fungi were serially diluted, plated on Sabouraud agar, and incubated for 24 h at 37 °C, and the CFU number was determined.

### 2.22. Magnetic Exposure Conditions

Experiments on the effects of the TR-system on the regulation of cell growth under different magnetic exposure conditions were performed with regular magnetic and shielded geomagnetic fields modulated by cultivating *C. albicans* in a custom-made box made of five layers of 10 µm-thick µ metal (to shield the geomagnetic field) at 37 °C for 6 h. Primary TezRs were inactivated on *C. albicans* as previously discussed. *C. albicans*, the final inoculum suspensions with an OD600 of 0.025 were inoculated into 4.0 mL of Sabouraud broth, and growth curves were generated via hourly measurements using a NanoDrop OneC spectrophotometer (Thermo Fisher Scientific, Waltham, MA, USA).

### 2.23. Light Exposure Experiments

For light irradiation, an aliquot of 4.0 log10 Vero cells was placed in the wells of 24-well plates. Cells were allowed to attach for 3.5 h at 37 °C in DMEM with 10% FBS, the medium was replaced with DMEM, and cells were treated with nucleases, as previously described to generate TezR–D1$^d$, TezR–R1$^d$, and TezR–D1$^d$/R1$^d$ cells. Cells were washed with DMEM, and fresh DMEM with FBS was added. The plates were exposed to visible light sources supplied with 150 W (840 lm) halogen lamps (Philips, Shanghai, China) for 24 h at 37 °C. The cellular state was observed and photographed under a Zeiss Axiovert 40C microscope (10× magnification).

### 2.24. In Vitro Antifungal Susceptibility Testing

Minimum fungicidal concentrations (MFCs) for RTIs and raltegravir were determined by microtiter assays in accordance with the Clinical and Laboratory Standards Institute (CLSI) guidelines [59]. A standard inoculum for *C. albicans* for testing was $2.5 \times 10^3$ CFU/mL. The tested compounds were serially diluted two-fold. The MFC was defined as the lowest concentration of antimicrobial agent that completely inhibited visible fungal growth after 48 h at 37 °C.

### 2.25. Effects of RT Inhibitors and Integrase Inhibitors on Cell Growth

The MFC against *C. albicans* was evaluated. *C. albicans* treated with vehicle and lacking primary TezRs was obtained as described previously. Fungi were incubated in Sabouraud broth supplemented or not supplemented with tenofovir and lamivudine as NRTIs, etravirine as an NNRTIs, abacavir as a nucleotide analog inhibitor, and raltegravir (5 µg/mL). Growth was monitored by measuring the optical density at OD600 during the first 12 h of growth and recorded at 1 h time intervals during the first 7 h and then at 12 h using a NanoDrop OneC spectrophotometer (Thermo Fisher Scientific, Waltham, MA, USA).

### 2.26. Biochemical Analysis

Biochemical tests were performed using the colorimetric reagent card YST card of the VITEK® 2 Compact 30 system (BioMérieux, France), according to the manufacturer's instructions. The generated data were analyzed using VITEK® 2 software version 7.01, according to the manufacturer's instructions.

### 2.27. Recognition of Maltose

The role of the TR-system in the recognition of maltose was investigated in *C. albicans*. Suspensions of control fungi and those following the destruction of primary TezRs were adjusted to a CFU incubated in fresh M9 medium supplemented or not supplemented with 146 mM maltose (Sigma-Aldrich). We measured the lag phase, which was taken as the period between the inoculation of fungi and that at which time the biomass began to grow, measured at OD600, which reflects the time required for the beginning of nutrient source utilization, using a NanoDrop OneC spectrophotometer (Thermo Fisher Scientific, Waltham, MA, USA).

### 2.28. Cell Memory Formation Experiments

We studied fungal memory formation based on the beginning of maltose utilization (time lag) for maltose-naïve and maltose-sentient *C. albicans*. To study the first exposure to maltose, *C. albicans* with unaltered TezRs was incubated in fresh M9 medium supplemented or not supplemented with 146 mM maltose (Sigma-Aldrich) for 24 h. To study the second exposure to maltose, fungi were taken after 24 h of cultivation from the first exposure to maltose, washed three times in PBS, and centrifuged at 4000× *g* for 15 min at 20 °C (Microfuge 20R, Beckman Coulter, Inc., Indianapolis, IN, USA). Fungi were adjusted to an 0.5 of OD600 and incubated in fresh M9 medium supplemented with maltose for the second time. During the first and second exposure to maltose, the samples were taken at 1 h

intervals for the first 7 h for OD600 analysis to determine the lag-phase with a NanoDrop OneC spectrophotometer (Thermo Fisher Scientific, Waltham, MA, USA). The difference in time lag between the first and second exposure to maltose was considered memory formation [60].

### 2.29. Evaluation of the Role of TR-System in Memory Formation

To study the role of TezR–R1 in remembering nutrient exposure, we assessed the difference in the minimal time required for TezR–R1 maltose-naïve and maltose-sentient *C. albicans* to trigger maltose utilization. Maltose-naïve and maltose-sentient *C. albicans* with unaltered TezRs were pretreated with water solutions containing 146 mM maltose (Sigma-Aldrich) for 5, 10, 30, 60, 90, or 120 min. Next, the fungi were treated with RNase, as previously discussed, to destroy TezR–R1 and inoculated in fresh M9 medium supplemented with maltose. We assessed the difference in the minimal time required for maltose-naïve and maltose-sentient fungi to begin maltose utilization by measuring the lag phase and monitoring the OD600 every hour using a NanoDrop OneC spectrophotometer (Thermo Fisher Scientific).

### 2.30. Memory Loss Experiments

The role of TezRs in fungal forgetting was studied based on the lag phase of maltose-naïve and maltose-sentient *C. albicans*. *C. albicans* with unaltered TezRs was cultivated in M9 medium supplemented with 146 mM maltose for 24 h, centrifuged at 4000× *g* for 20 min (Microfuge 20R, Beckman Coulter, Inc., Indianapolis, IN, USA), and washed in M9 media without maltose. The cells were then subjected to repeated rounds of TezR–D1/R1 destruction and restoration. TezR–D1/R1 were destroyed, as previously described, and the fungi were inoculated into M9 broth without maltose. After 24 h of cultivation at 37 °C, fungi were isolated from the media, TezR–D1/R1 was destroyed again, and fungi were again inoculated in fresh nutrient broth. Three sets of cultivations in broth followed by TezR–D1/R1 destruction were performed. After each set of cultivation in nutrient broth, prior to TezR–D1/R1 destruction, some part of the probe was removed, fungi were washed out and inoculated in M9 broth supplemented with maltose, and the time lag was assessed based on the OD600. After the third round of cultivation in M9 broth, fungi were centrifuged and inoculated into fresh M9 broth without the destruction of TezR–D1/R1, cultivated for 24 h, centrifuged, washed out from the media, and inoculated into M9 broth supplemented with maltose (modulating the second contact with maltose), and the time lag was assessed by OD600. Fungi from the control group were processed in the same way, but treated with water.

### 2.31. Raltegravir in Cell Memory Formation Experiments

To study the effect of raltegravir on cell memory, *C. albicans* was grown on fresh M9 medium supplemented or not supplemented with 146 mM maltose, as previously described, with or without additional supplementation with raltegravir (5 μg/mL). To evaluate the maximal time during which raltegravir affects maltose utilization, *C. albicans* was grown in M9 broth supplemented with 146 mM maltose, and raltegravir was added at 0 h, 15 min, 30 min, 60 min, or 120 min of growth. To determine the lag phase, samples were taken at 1 h intervals for the first 7 h for OD600 analysis with a NanoDrop OneC spectrophotometer (Thermo Fisher Scientific, Waltham, MA, USA).

### 2.32. Seedling Growth Assay

Seeds of *Triticum aestivum* were surface sterilized with 1% NaClO for 10 min, then rinsed five times with distilled water. Seeds were treated with a solution of DNase, RNase, or a combination thereof taken at 50 μg/mL for 30 min, rinsed five times with distilled water, and germinated for 1 d in the dark on floating plastic nets. Control seeds were processed in the same manner but were treated with water instead. After germination, seedlings were sown in L 2-L-pots containing Farfard professional potting mix (Conrad

Farfard Inc., Agawam, MA, USA) with a density of three seeds/pot. The pots were grown in a controlled environment, the Conviron growth chamber, as previously described [61].

The chlorophyll content of leaves was determined 7 d after seed placement, according to He et al., with some modifications. Fresh leaf material (50 mg) was homogenized in 10 mL of 95% ethanol. The homogenate was centrifuged at $1500 \times g$ for 20 min, and the supernatant was collected, and measured using a NanoDrop OneC spectrophotometer (Thermo Fisher Scientific, Waltham, MA, USA) at 649 and 665 nm. The concentrations of chlorophyll-$\alpha$, chlorophyll-$\beta$, and total chlorophyll ($\alpha + \beta$) were calculated using the equations described by Porra and Liu [62,63]. The total chlorophyll content was determined using the following formula:

$$\text{chlorophyll-}\alpha = 13.95 \times \text{A665} - 6.68 \times \text{A649} \tag{1}$$

$$\text{chlorophyll-}\beta = 24.96 \times \text{A649} - 7.32 \times \text{A649} \tag{2}$$

$$\text{Total chlorophyll} = (\text{chlorophyll-}\alpha + \text{chlorophyll-}\beta) \times \text{final volume of sample (mL)} \times \text{dilution fold/fresh weight of sample taken} \tag{3}$$

was expressed as mg chlorophyll g$-1$ fresh weight by using the following equation:

$$\text{Total chlorophyll (mg g}^{-1}\text{ FW)} = [20.2(\text{D645}) + 8.02(\text{D663})] \times [V/(1000 \times W)],$$

where V = volume of 80% aqueous acetone (mL), W = weight of fresh leaf (g), D645 = absorbance at 645 nm wavelength, and D663 = absorbance at 663 nm wavelength.

### 2.33. Generation of RNA Sequencing Data

To isolate RNA from Vero cells, the cell suspension obtained 2.5 h post-nuclease treatment were washed thrice in PBS, pH 7.2 (Sigma) and centrifuged each time at $4000 \times g$ for 20 min (Microfuge 20R, Beckman Coulter) followed by resuspension in PBS.

RNA was purified using the RNeasy Mini Kit (Qiagen), according to the manufacturer's protocol. The concentration and quality of RNA based on absorbance at 230, 260, and 280 nm was determined with the NanoDrop OneC spectrophotometer (Thermo Fisher Scientific).

Transcriptome sequencing (RNA-Seq) libraries were prepared using an Illumina TruSeq Stranded Total RNA Library Prep kit. RNA was ribo-depleted with the Epicenter Ribo-Zero magnetic gold kit (catalog No. RZE1224), according to the manufacturer's recommendations. The libraries were pooled equimolarly and sequenced in an Illumina NextSeq 500 (Illumona, San Diego, CA, USA) platform with paired 150-nucleotide reads (130MM reads max).

### 2.34. Analysis of RNA Sequencing Data

Sequencing reads were mapped corresponding to the reference genome of *S. aureus* NCTC 8325 (NCBI Reference Sequence: GCA–015252025.1), and expression levels were estimated using Geneious 11.1.5. Transcripts with an adjusted $p$ value of <0.05 and $\log_2$ fold change value of $\pm 0.5$ were considered for significant differential expression. PCA, volcano plots were generated using the ggplot2 package in R, and the Venn diagram was obtained using BioVenn [64]. Differentially expressed genes (DEGs) were identified as genes with a two-fold change ($\log_2$ fold-change > 0.5 or <$-0.5$) and false discovery rate < 0.05.

### 2.35. Statistics

At least three biological replicates were performed for each experimental condition unless stated differently. Each data point was denoted as the mean value $\pm$ standard deviation. A two-tailed $t$-test was performed for pairwise comparisons, and a $p$-value $\leq 0.05$ was considered significant. Fungal quantification data were log10 transformed prior to analysis. Statistical analyses for the biofilm assays and hemolysin tests were performed using the Student's $t$-test. Data from animal and sporulation studies were calculated using

the two-tailed Mann–Whitney U test. GraphPad Prism version 9 (GraphPad Software, CA, USA) or Excel 10 was used for statistical analyses and illustrations.

## 3. Results

### 3.1. Classification and Nomenclature of TezRs

TezRs have been further classified based on the structural features of the domain located outside the membrane and their association with the cell surface, including the possibility of being washed with culture media (Table 1).

**Table 1.** Classification of TezRs in eukaryotes.

| Name of the Receptor | Description of the Receptor |
|---|---|
| Primary TezRs | |
| TezR–D1 | DNA-based receptors located outside the membrane; stably associated with the cell surface. |
| TezR–R1 | RNA-based receptors located outside the membrane; stably associated with the cell surface. |
| Secondary TezRs | |
| TezR–D2 | DNA-based receptors located outside the membrane; can be easily washed out along with culture medium or matrix. |
| TezR–R2 | RNA-based receptors located outside the membrane; can be easily washed out along with culture medium or matrix. |

To describe cells in which certain TezRs were deactivated, we marked them with the superscript letters "d", meaning deactivated. As an example, Vero cells with deactivated primary DNA-based TezR are designated as "Vero TezR-D1$^{d}$", where TezR stands for receptor and is followed by a dash, and then a capital letter representing the type of nucleic acid (D for DNA), followed by an Arabic numeral indicating that it is a primary receptor, and superscript "d" meaning that this receptor was deactivated. The same principle of naming was applicable to cells with other destroyed TezRs. Cells with multiple cycles of TezR destruction and restoration were named "Zero cells" and are designated by a superscript letters "zdr".

We confirmed the presence of cell surface-bound nucleic acid based TezRs through the analysis of altered fluorescence in Vero or CHO cells following the removal of the extracellular matrix and their treatment with treated with water (control) or a combination of DNase I and RNase at a concentration of 10 μg/mL for 10 min. Control PI stained cells displayed clear fluorescence around the cells, and those stained with Syto 9 revealed fluorescence on the cell surface as well as nucleus, confirming the presence of cell surface-bound nucleic acids, which remained after the removal of culture medium (Supplementary Figure S1A,B).

For both cell lines, cells treated with either DNase or RNase alone exhibited a decrease in fluorescence around the cells compared to vehicle-treated cells. However, cells treated with a combination of DNase and RNase displayed the total disappearance of surface fluorescence.

### 3.2. Cell Cycle and Apoptosis Regulation by TR-System in Mammalian Cells

To detect whether the loss of primary TezRs induces apoptosis in mammalian cells, we used Vero and HEK293 cells. We observed cellular and nuclear morphological changes using YO-PRO-1 and propidium iodide (PI) staining. The viability of Vero cells was not affected by TezR inactivation (Figure 1A,C). An increase in the percentage of early and late apoptotic cells was observed with a decrease in TezR–D1, TezR–R1, and TezD1/R1 in HEK293 cells ($p < 0.01$; Figure 1B,D). From these results, we concluded that the loss of primary TezRs differentially affected early- and late-stage apoptosis in different cell lines.

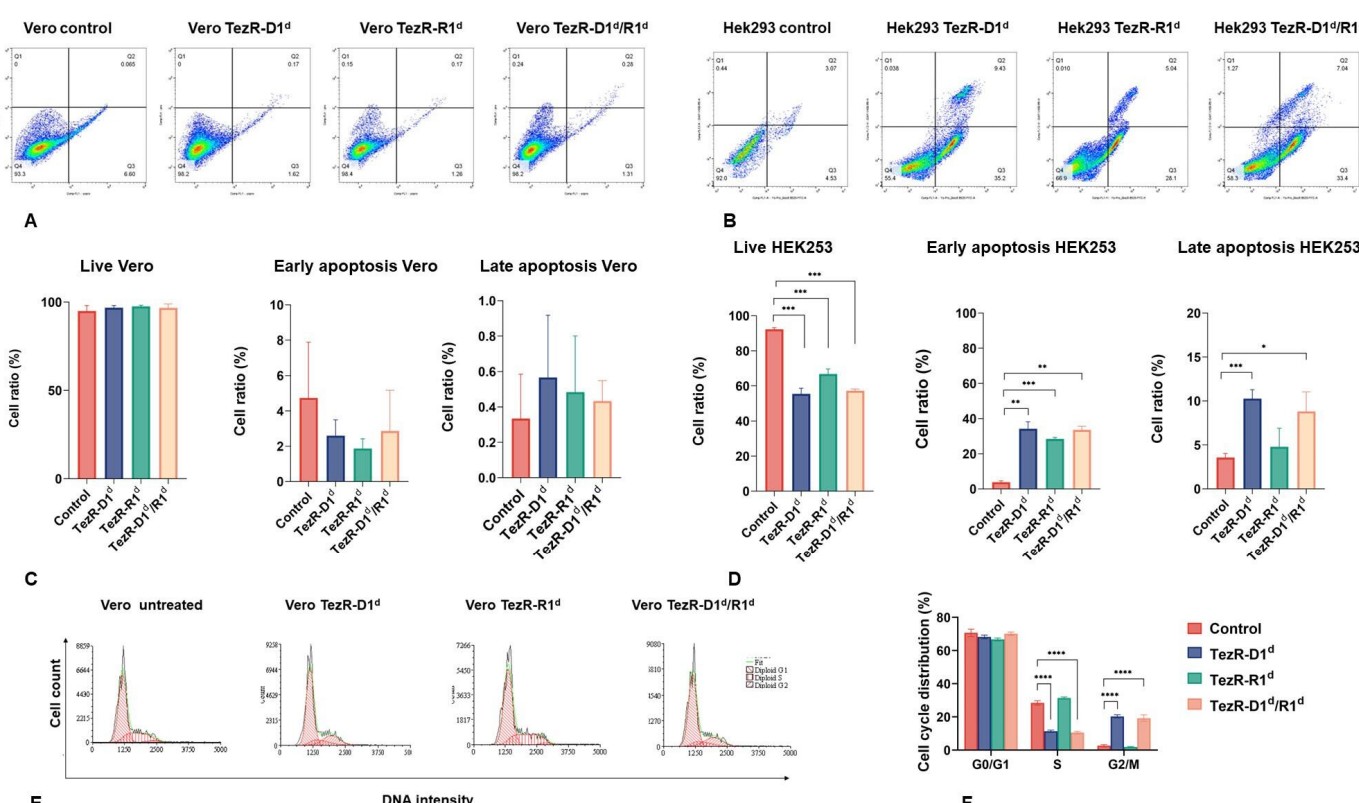

**Figure 1.** Effects of TezR inactivation on mammalian cells. Scatter plots of (**A**) Vero cells and (**B**) HEK293 stained with YO-PRO1/PI (propidium iodide) under four situations in quadrant analysis; viable cells (Q4), early apoptotic cells (Q3), late apoptotic cells (Q2), and necrotic cells (Q1) are shown. All cells were treated with water (control), DNase and/or RNase to destroy TezR–D1, TezR–D2, and TezR–D1/R1 and were allowed to grow for 3 h. Cells were treated with the vehicle as a control. Similar plots were observed in three independent experiments (*n* = 3). (**C,D**) The percentage of live cells (left panel), cells in early apoptosis (middle panel), cells in late apoptosis, or dead cells (right panel) among (**C**) Vero and (**D**) HEK293 cells. Triplicate samples were used to obtain the mean and standard deviation. The asterisks indicate statistical significance. **** $p < 0.001$, *** $p \leq 0.001$, ** $p \leq 0.01$, * $p < 0.05$; (**E**) Vero cells following primary TezR destruction were subjected to cell-cycle analysis using flow cytometry. The representative histogram of Vero TezR–D1$^d$, TezR–R1$^d$, and TezR–D1$^d$/R1$^d$ cells revealed an acceleration of S-phase after the loss of primary DNA-formed TezRs compared with water-treated control. (**F**) A quantitative analysis of the distribution or proportion of cells in each phase was performed from at least 10,000 cells per sample. Each bar represents the mean ± SD of the data obtained from three independent experiments. **** $p < 0.001$.

The effects of primary TezR loss on cell cycle phases were analyzed using flow cytometry in Vero cells (Figure 1E). Quantitative data revealed that the destruction of TezR–D1 or TezR–D1/R1 accelerated S phase progression (Figure 1F). Thus, Vero TezR–D1$^d$ cells had an approximate 2.4-fold reduced distribution in S phase and 7.6-fold increased distribution in G2 phase ($p < 0.0001$). Similarly, Vero TezR–D1$^d$/R1$^d$ cells showed a 2.7-fold lesser proportion in S phase with a 7.2-fold increase in G2 phase ($p < 0.0001$). In both cases, the number of cells in the G1 phase continued to remain the same as that seen in controls. These results show that primary DNA-based TezRs are implicated in cell cycle.

### 3.3. TezRs Regulate Mammalian Cell Morphology and Migration

Next, we examined the role of TezRs in mammalian cell growth and plating using Vero cells whose viability was not affected by the loss of primary TezRs. We also confirmed that RNase A in the settings used was not internalized by the cells. We studied the penetration

of RNase A linked with a fluorophore in living Vero cells treated with RNase, but no signs of RNase internalization were observed. (Supplementary Figure S2).

Control cells and those after the destruction of primary TezRs grew and formed confluent monolayers within 48 h with a regular morphology. (Figure 2A). The destruction of TezR–R1 resulted in significantly decreased cell sizes ($p < 0.001$) in cells after 48 h, whereas the individual destruction of TezR–D1 or combined destruction of TezR–D1/R1 had no significant effect on cell size measurements (Figure 2B).

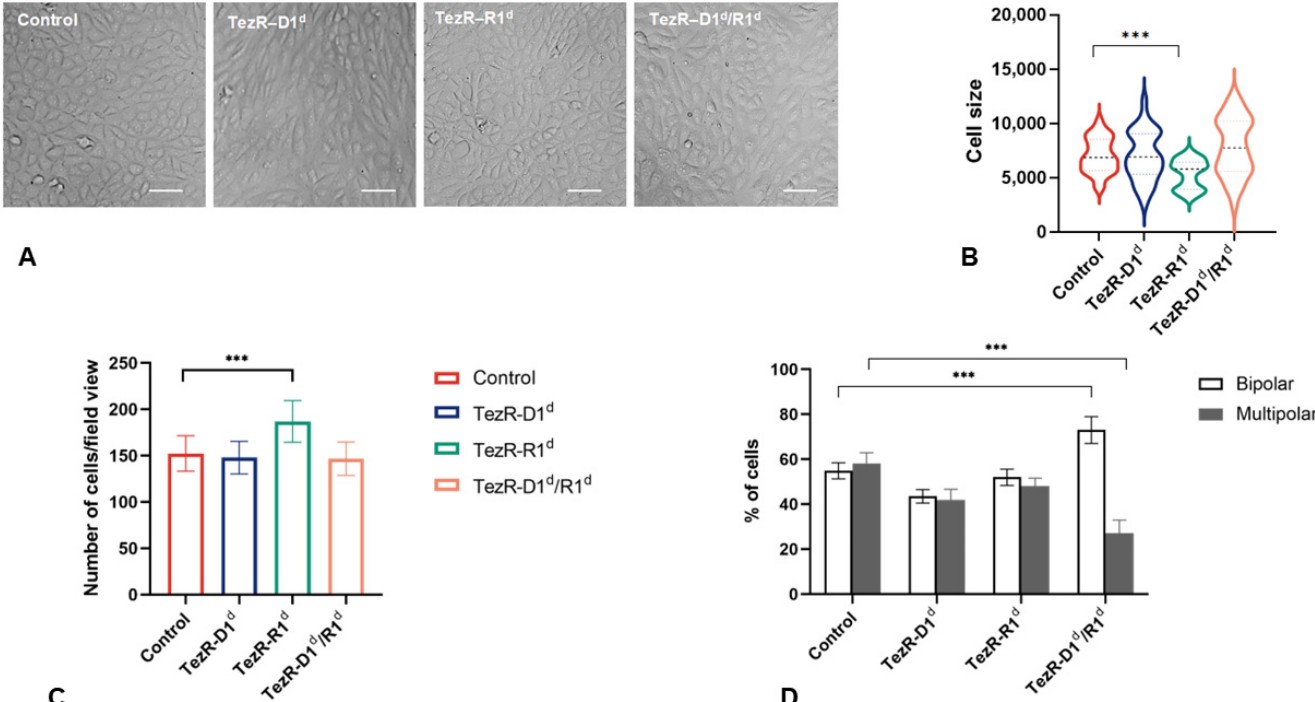

**Figure 2.** Effects of TezR destruction on mammalian cell morphology. (**A**) The morphology of Vero TezR–D1$^d$, Vero TezR–R1$^d$, and Vero cells TezR–D1$^d$/R1$^d$ following treatment with nucleases, as discussed in the Materials and Methods, was studied by light microscopy. Cells were treated with the vehicle as a negative control. Similar cellular morphology was observed in five independent experiments. Light microscopy, ×10. The scale bars represent 100 μm. (**B**) Cell area and images in control and Vero cells and those following TezR destruction after culture on the dish for 48 h after destruction. Data are the average ± SD of the cell area. *** $p < 0.001$. (**C**) The number of attached cells was increased following TezR destruction in 24 h-old Vero cell cultures. Error bars represent the standard deviation from three independent experiments. *** $p < 0.001$. (**D**) TezR destruction affects cell morphology. Proportion of bipolar or multipolar cells following destruction of different TezRs. Data represent the averages of three independent experiments and were compared with the water-treated control. Error bars represent standard deviation. *** $p < 0.001$.

Microscopy of subconfluent 24 h-old cell cultures after the destruction of primary TezRs showed that the number of cells was increased following TezR–R1 destruction compared to that with controls ($p < 0.001$; Figure 2C; Supplementary Figure S3). As shown in Figure 3D, the proportions of multipolar morphotypes vs. bipolar morphotypes were different in 24 h-old cell cultures depending on the destruction of TezRs. The percentage of multipolar cells decreased from 43.5% to 27% ($p < 0.001$) in cells following TezR–D1/R1 destruction compared to that in controls, and the proportion of fusiform morphotypes increased (Figure 2D; Supplementary Figure S3).

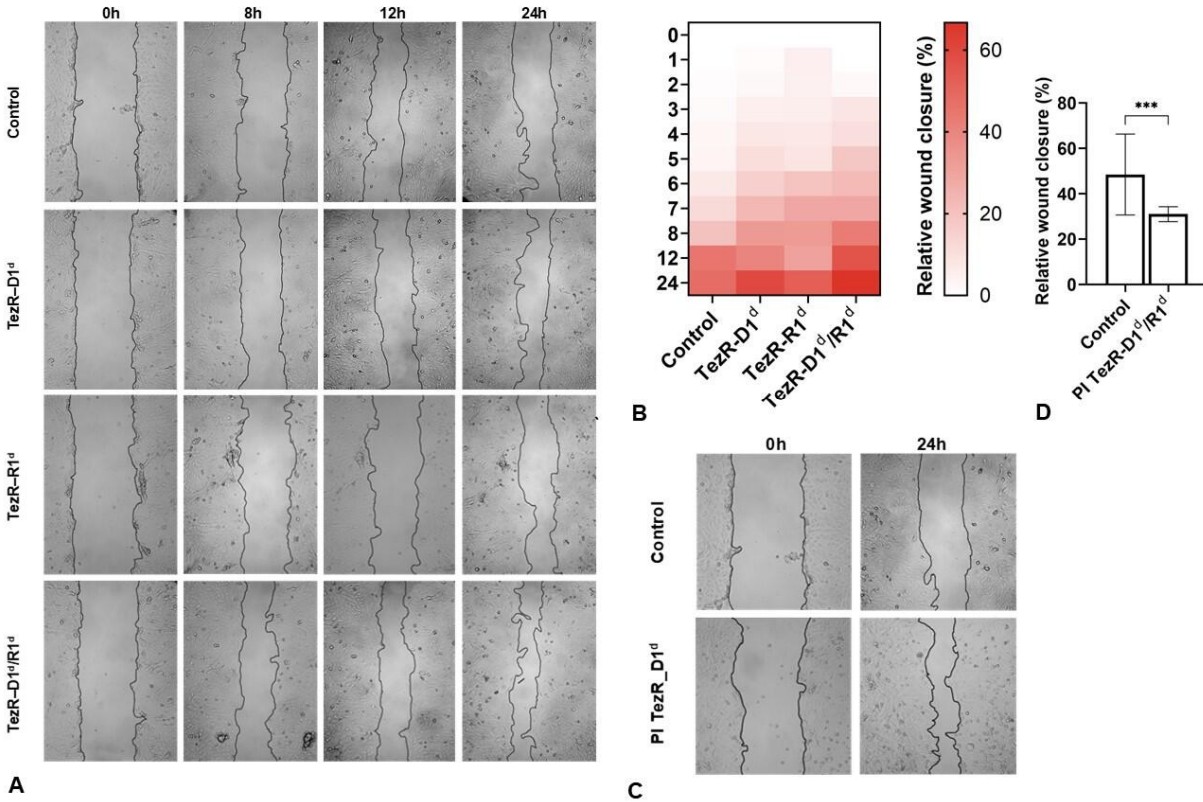

**Figure 3.** TezRs regulate cell migration. The effects of primary TezR loss or inactivation with PI was compared with water-treated control cells. (**A**) Individual snapshots at different times (from 0 to 24 h) after the start of the assay showing gap filling by Vero cells after the destruction of different primary TezRs. The frontline boundary of the moving cells is outlined in black. The destruction of TezR–D1/R1 accelerated the migration of Vero cells at all measured timepoints, specifically 8, 12, and 24 h. The destruction of TezR–D1 accelerated the migration of Vero TezR–D1$^d$ cells only within 8 h of observation time. (**B**) Heatmap showing the differences in migration rates. The color intensity of percentage wound closure is represented by a color scale, from white (minimal) to red (maximum). Data represent the average of three independent experiments. We observed that Vero TezR–D1$^d$, TezR–R1$^d$, and TezR–D1$^d$/R1$^d$ cells started moving into the gap earlier than the control cells, and the earliest migration was noted for Vero TezR–R1$^d$ cells, which started moving after 1 h. All results are representative of three independent experiments; *** $p \leq 0.001$. (**C**) Gap filling by Vero cells after the binding of TezR–D1/R1 with PI. The frontline boundary of the moving cells is outlined in black. This inactivation accelerated the migration of Vero cells. (**D**) Values of percentage wound closure $\pm$ SD ($n = 3$); *** $p = 0.001$.

We also investigated whether the morphological changes elicited by TezR inactivation could contribute to cell migration using a wound-healing assay and time-lapse microscopy [65]. In terms of gap-filling, in minimal starvation media that allowed cells to migrate, we first noted that the destruction of TezRs induced faster wound closure, but the kinetics of wound closure were dependent on the type of TezR destroyed and the observational time (Figure 3A).

When migration rate kinetics in the early stages of wound closure were studied, it was observed that cells with destroyed TezRs started moving into the gap much earlier than control, water-treated cells (Figure 3B). The short-term time-lapse analysis demonstrated the earliest cell migration starting from 1 h for Vero–TezR–R1$^d$ cells, followed by 3 h for TezR–D1$^d$ and TezR–D1$^d$/R1$^d$ cells, compared with 5 h observed for the control. Over 8–24 h, Vero TezR–D1$^d$ cells exhibited accelerated kinetics compared to the controls within 8 h of observation time, but this effect was diminished during subsequent growth. The

destruction of TezR–R1 did not lead to any statistically significant acceleration of wound closure within the same timeframe.

The fastest kinetics were observed for Vero TezR–D1$^d$/R1$^d$ cells with 43%, 56%, and 72% wound closure at 8, 12, and 24 h of growth, respectively, compared with those in water-treated cells, which displayed 20%, 45%, and 48% closure, respectively, at the same time points ($p < 0.05$). We also used PI, thus suggesting that it altered the function of TezRs via binding. Treatment with PI significantly accelerated the kinetics of wound closure reflected as the "relative wound closure" area (by more than 45%; $p = 0.001$) compared with those in the controls, with results similar to those obtained following the destruction of primary DNA- and RNA-based TezRs and the generation of TezR–D1$^d$/R1$^d$ (Figure 3C,D). These observations indicate that TezRs act as critical signal regulators in the migration of mammalian cells.

### 3.4. TezRs Regulate Mammalian Gene Expression

RNA-seq was performed to profile the gene expression of Vero cells following destruction of primary TezRs.

To address the differences in transcriptome profile between Vero cells following the loss of any primary TezRs and control cells, principal component analysis (PCA) was first used to study potential clusters based on differently detected genes. The PCA plot visually showed that PC1 and 2 separated the Vero control, TezR-D1$^d$, TezR-R$^d$ and TezR-D1$^d$/R1$^d$ as four distinctive clusters (A).

Although the largest difference in PCA was observed in cells with destroyed TezR–R1$^d$, this PCA result indicated that Vero cells following the loss of any primary TezRs could be separated by their transcriptome profile by clustering.

Next, we compared the results from each probe and analyzed the genes for which expression was significantly altered (upregulated or downregulated) following the removal of different TezRs. Volcano plots of the log2(fold change) of DEGs ($|\log_2$ fold-change$| > 0.5$ and $p$-value $< 0.05$) show that many of the differentially upregulated and downregulated changes were highly significant in Vero cells lacking TezRs compared to control cells (Figure 4B–D, Supplementary Table S1). We observed individual responses following the loss of different TezRs, highlighting individual regulatory roles of TezRs. Comparing DEGs between control Vero cells and cells lacking TezRs, we found 70 upregulated and 136 downregulated genes with TezR–D1$^d$, 40 upregulated and 72 downregulated genes with TezR–R1$^d$, and 69 upregulated and 80 downregulated genes with TezR–D1$^d$/R1$^d$ (Figure 4E). Moreover, cells following the loss of both TezR–D1 and TezR–R1 had an individual and complex response, which cannot be justified by summing up the effects of individual TezRs loss.

An analysis of the top 10 altered genes following the loss of primary TezRs revealed interesting results (Figure 4F–H). After the destruction of TezR–D1, genes involved in the positive regulation of apoptotic processes (*DAPK2*), ATP production (*MT-ND2*, *ND1*), mitotic G2 checkpoint signaling (*FOXN3*), and positive regulation of transcription (*SNAI1*) were upregulated, whereas the number of genes associated with the positive regulation of protein catabolic processes (*MYLIP*), cell proliferation (*WNT3A*), and those associated with the regulation of membrane potential (*ZACN*) were downregulated (Figure 4F). Among the upregulated genes in cells following the inactivation of TezR–R1, they were primarily related to sodium ion transmembrane transport (*SLC6A15*), ATP production (*ND4L*, *ND1*, *ND5*, *MT-ND4*, *MT-ND2*, *ATP6*), and the negative regulation of transcription (*FOXN3*). Under the same conditions, genes involved in the positive regulation of chemotaxis (*CXCR4*, *CXCR3*), the regulation of chloride and sodium transmembrane transport (*FXYD4*, *CLCN1*), and signaling, including protein kinase A signaling (*SPATC1L*, *GHRH*), were downregulated (Figure 4G).

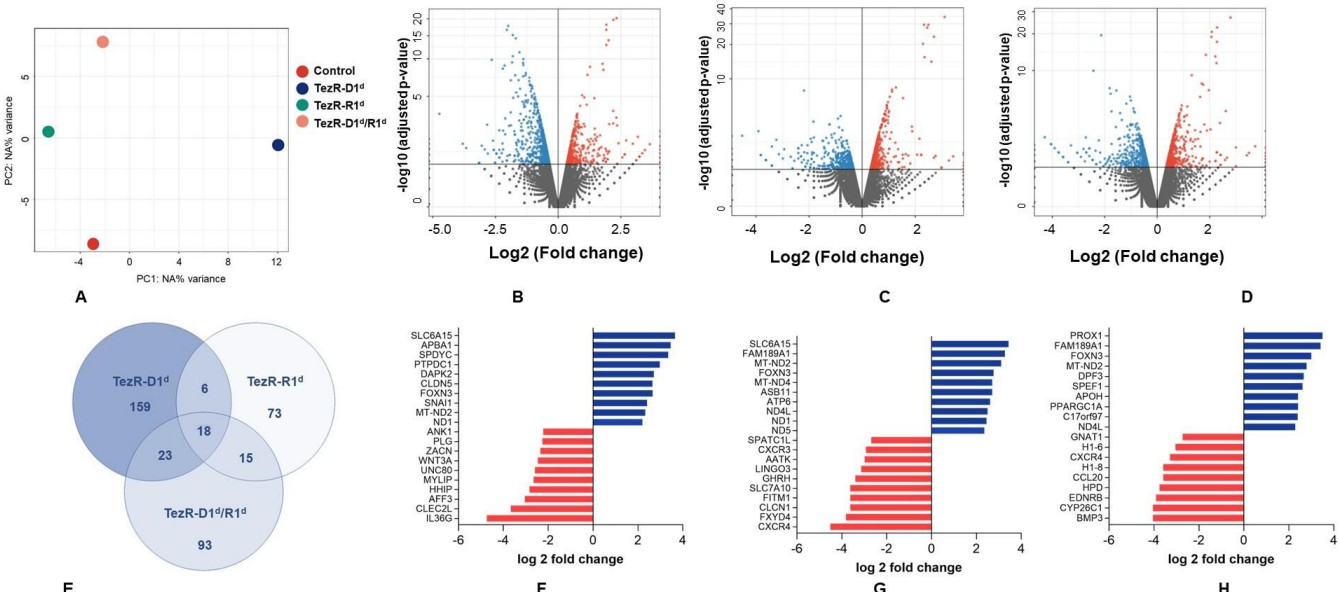

**Figure 4.** Comparison of transcriptome profiles of Vero cells following the loss of primary TezRs (**A**) Principal-component analysis (PCA) plot (**B–D**) and volcano plots demonstrating gene expression changes (log$_2$ fold-change > 0.5 change plotted against the $-$log$_{10}$ P-value). (**E**) A Venn diagram was created to compare differentially expressed genes (DEGs) with increased and decreased abundance between following the loss of primary TezRs. (**F**) Expression of top 10 highly upregulated or downregulated TezR–D1-dependent genes (all with $p < 0.05$). (**G**) Expression of top 10 highly upregulated or downregulated TezR–R1-dependent genes (all with $p < 0.05$). (**H**) Expression of top 10 highly upregulated or downregulated TezR–D1/R1-dependent genes (all with $p < 0.05$).

In the cells lacking TezR–D1/R1, the upregulation of genes involved in the regulation of gene expression (*PROX1*), cell migration (*SPEF1*), negative regulation of transcription and cell proliferation (*APOH*, *DPF3*), response to oxidative stress (*PPARGC1A*), and ATP production (*ND4L*, *MT-ND4*) was observed. The loss of TezR–D1/R1 resulted in the downregulation and translational repression of genes implicated in calcium ion transport (*WNT3A*), cell surface receptor signaling pathways (*EDNRB*, *CCL20*, *GNAT1*, *BMP3*), apoptosis (*CXCR4*), and cell differentiation (H1–6) (Figure 4H). Intriguingly, the only group of genes for which expression was significantly upregulated due to the loss of any of the primary TezRs comprised those associated with mitochondrial electron transport and ATP production. These results suggest that primary TezRs are important elements of cells and their inactivation results in multiple alterations of genes expression.

### 3.5. Role of TezRs in Fungal Growth and Viability

The effects of TezRs on *Candida albicans* viability were assessed by spectrophotometry and the enumeration of colony-forming units (CFUs). Stationary-phase *C. albicans* were treated with nucleases to destroy primary TezRs, after which, nucleases were washed out, and cells were allowed to grow in fresh media. The control probes were treated with the vehicle. The OD$_{600}$ and CFU counts were measured hourly from 0 to 7 h and at hour 12 of incubation (Figure 5A,B).

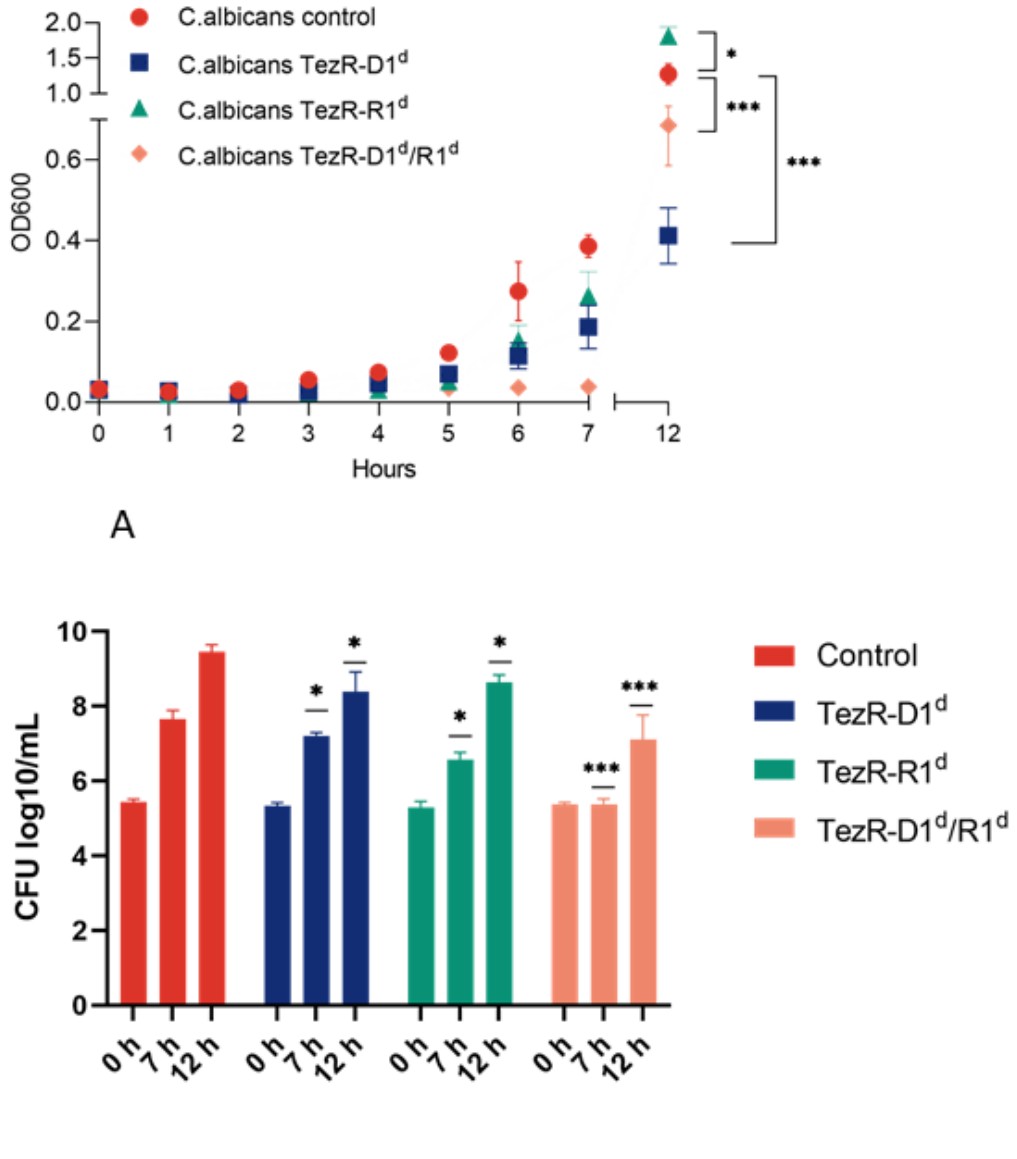

**Figure 5.** Effect of TezR destruction on *Candida albicans* viability. (**A**) A comparison of control fungi and fungi with destroyed TezRs revealed retardation of growth with an increase in the lag phase following the loss of any primary TezRs. Error bars represent the standard deviation from three independent experiments. * $p < 0.05$, *** $p < 0.0001$. (**B**) Fungal growth measured as fungal counts ($\log_{10}[\text{CFU/mL}]$) after TezR destruction showed the delay of *C. albicans* TezR–D1[d], TezR–R1[d], and TezR–D1[d]/R1[d] growth within the first 12 h. The columns represent averages and error bars represent standard deviations from three independent experiments. * $p < 0.05$, *** $p < 0.001$ as compared to the control *C. albicans* taken at the same time.

The inactivation of any primary TezRs resulted in growth retardation with an extended time lag compared with that in vehicle-treated *C. albicans*, as measured by the OD600 within the first 7 h. However, only the removal of TezR–D1 or TezR–D1/R1 still inhibited fungal growth by 12 h (at the end of the observation period), whereas the OD600 of *C. albicans* TezR–D1[d] even exceeded that of the control. We observed the suppression of fungal growth, expressed as CFUs, following the destruction of primary TezRs when measured after 12 h. It is reasonable to assume that delayed regrowth might depend on the time the fungal

cells need to restore physiological functions after non-lethal damage due to the destruction of TezRs.

### 3.6. Dependence of Fungal Cell Morphology and Biochemical Characteristics on TezRs

The effects of TezR inactivation on fungal morphology were studied in *C. albicans*. No signs of RNase A internalization were observed when the fungi were either treated with RNase or cultivated on agar supplemented with RNase (Supplementary Figure S4). The percentages of the different morphological forms were quantified using the morphological index (MI) of individual cells after 72 h of growth at 37 °C compared with water treated control (Figure 6A,B) [66]. Values close to 1 indicate a population of spheroidal yeast cells, and values close to 5 indicate a population of true hyphal cells. The MI for water treated cells revealed mixed yeast, pseudohyphal, and hyphal morphologies. The loss of TezR–D1 or TezR–R1 inhibited both filamentous pseudohyphal and hyphal formation, which was slightly less inhibited after the inactivation of primary RNA-based TezRs. In contrast, after the combined destruction of TezR–D1/R1, cells exhibited a diverse MI, as with control cells, but true hyphae formation was evident in more than 35% of the population, which was higher than that of the control.

Owing to the observed discrepancy in *C. albicans* TezR–D1$^d$/R1$^d$, which despite lacking TezR–D1 and TezR–R1 (the destruction of which inhibited pseudohyphal and hyphal formation) had MI values similar to those of control fungi, we named these cells "Drunk cells". Studying the role of TezRs in the regulation of fungal cell size, we found that the loss of TezR–R1 slightly decreased cell size, but this trend was not statistically significant (Figure 6C). We also studied alterations in the biochemical profile of *C. albicans* following primary TezR destruction with Vitek 2, enabling the study of their role in regulating the assimilation of different compounds [67]. The inactivation of primary TezRs altered the biochemical activity of the fungi. The profiles of activated and inactivated cell enzymes were different following the loss of different TezRs (Figure 6D). The destruction of TezR–D1 led to the inactivation of five enzymes compared to that in the control. One of the key alterations after the removal of TezR–D1 was the inhibition of leucine-arylamidase, a hydrolytic enzyme that plays an important role in the colonization and invasion of *C. albicans* into the host tissue [68]. We also found a switch to a negative reaction for α-glucosidase, which is known for its role in overall cell fitness, cell wall assembly, and virulence in *C. albicans* [69,70]. Two enzymes that were activated after TezR–D1 destruction were those implicated in erythritol and gentiobiose assimilation, which can be used as sugar substitutes [71].

*C. albicans* TezR–R1$^d$ was negative for nine enzymes and active for three enzymes compared with those in the control. Interestingly, the destruction of TezR–R1$^d$ activated assimilation of the di- and trisaccharides maltose and raffinose, but inhibited glucose assimilation, thereby affecting the glyoxylate cycle, which is a part of the central metabolism of *C. albicans* [72]. Similar to that in *C. albicans* TezR–D1$^d$, following the destruction of TezR–R1, α-glucosidase activity was also inhibited.

The most profound inhibitory effect was observed in *C. albicans* TezR–D1$^d$/R1$^d$ "Drunk cells", which resulted in the inactivation of 13 enzymes and activation of six. These changes included alterations in typical enzyme functions in *C. albicans* TezR–D1$^d$ and TezR–R1$^d$, as well as individual signatures, such as inhibition of xylitol and sorbitol, amygdalan, and D-galactose assimilation, whereas urease and lactose assimilation were activated. The development of a positive reaction for urease is not typical for *C. albicans*, which lacks urease and uses the cytoplasmic enzyme urea amidolyase to assimilate urea [73,74].

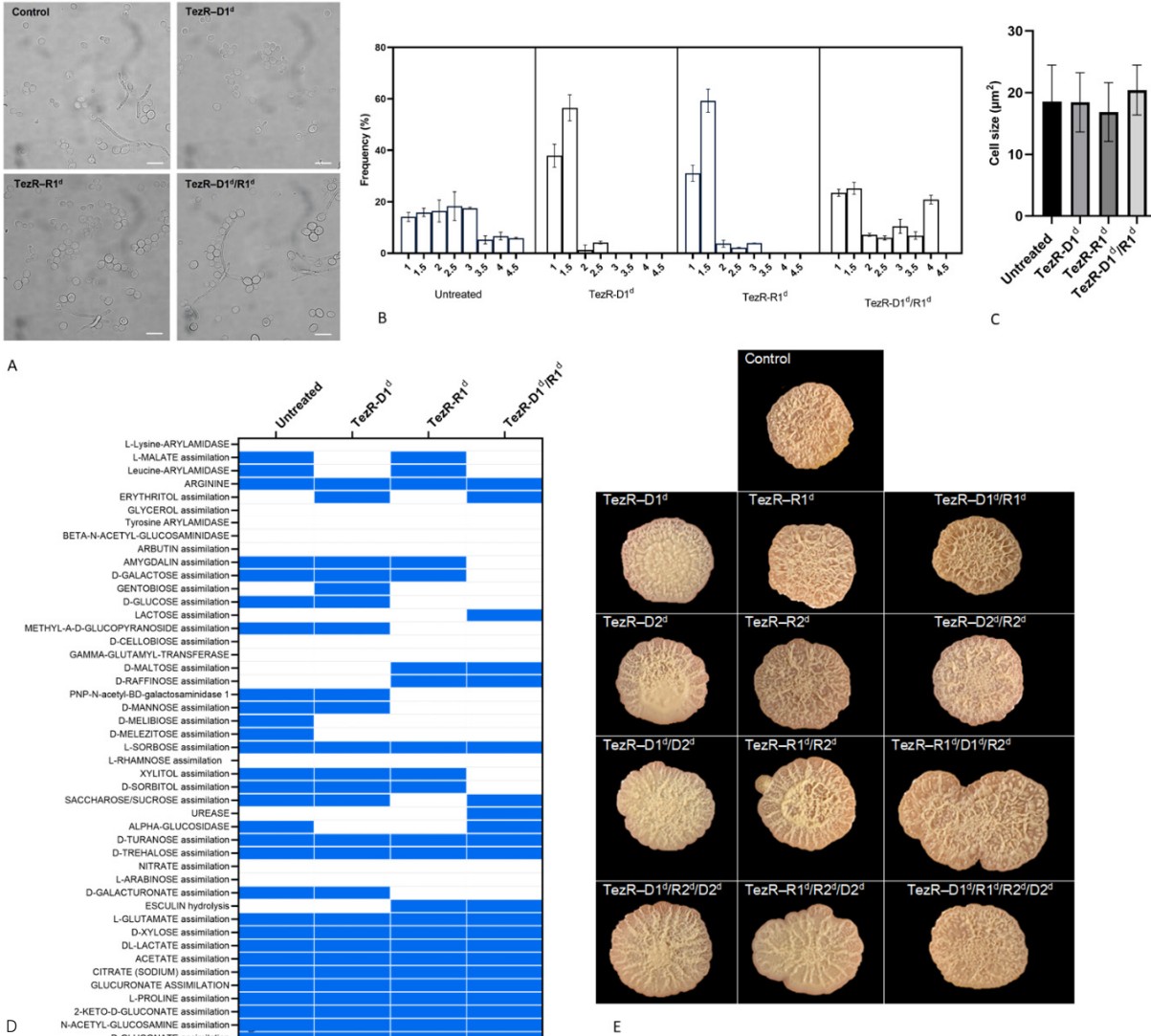

**Figure 6.** Effects of TezR destruction on Candida albicans hyphal growth on solid media. (**A**) The destruction of primary TezRs affected the mode of fungal growth. The destruction of TezR–D1 and TezR–R1 inhibited hyphal and pseudohyphal formation. Aliquots were withdrawn and photographed using a bright-field microscope. The scale bars represent 10 μm. (**B**) The morphological indices (MI) of yeast, pseudohyphal, and hyphal cells were measured. Error bars are ±SD, with averages from 10 image measurements from three replicates. (**C**) The fungal cell area (μm$^2$) was analyzed for a total of 500 yeast cells. Data are presented as the mean ± SD from three independent experiments. (**D**) The biochemical characteristics of *C. albicans* following the destruction of primary TezRs were studied using a Vitek-2 system. Test reaction data are shown as "positive," marked with a blue color or "negative," marked with white color. Data are representative of three independent experiments. (**E**) Morphology of *C. albicans* biofilm following TezR destruction. To study the role of primary TezRs in biofilm morphology, 25 μL of *C. albicans* with destroyed primary TezRs (*C. alibcans* TezR–D1$^d$, TezR–R1$^d$, TezR–D1$^d$/R1$^d$) was inoculated into the center of dry agar plates without nucleases. To study the effects of secondary TezRs, untreated *C. albicans* cells were plated on agar plates supplemented with DNase to generate *C. alibcans* TezR–D2$^d$, RNAse to generate *C. alibcans* TezR–R2$^d$, and DNase and RNAse to generate *C. albicans* TezR–D2$^d$/R2$^d$. To study the combined effects of primary and secondary TezR loss on biofilm morphology, *C. alibcans* pretreated with nucleases to destroy primary TezRs was inoculated into agar supplemented with nucleases to destroy secondary TezRs. The morphology of the fungal colonies was photographed after incubation for 7 days at 37 °C and then photographed. Images are representative of three independent experiments.

The activation of enzymes required for the assimilation of lactose by *C. albicans* following TezR–D1/R1 destruction was even more surprising, since normally, among *Candida* spp., lactose is assimilated only by *C. kefyr,* and this fact is used to differentiate genera and species of yeasts [75,76].

Next, we found that *Candida* biofilm morphology was influenced by TezRs. To analyze the role of primary TezRs, *C. albicans* was pretreated with nucleases and then inoculated and grown on Sabouraud agar medium. To study the role of secondary TezRs, the growth of *C. albicans* was evaluated on Sabouraud agar supplemented with nucleases. The results indicated that the biofilm morphology of *C. albicans* was significantly influenced by primary and secondary TezRs. The destruction of certain types of TezRs alone or the combined destruction of multiple types of TezRs affected the thickness of the biofilm and wrinkle morphology, promoting or decreasing the height of the wrinkles, the formation of symmetrical radial wrinkles or circular rings, and the thickness and morphology of the edge. In particular, biofilms of *C. albicans* TezR–R1$^d$/R2$^d$, with the combined destruction of primary and secondary RNA-formed TezR, exhibited biofilm dispersal with the formation of novel small biofilms. Another unique result was noted for biofilms formed by TezR–R1$^d$/D1$^d$/R2$^d$ cells with the formation of irregularly spread biofilms.

In these experiments, nucleases were added to the solid nutrient medium in an attempt to destroy fungal secondary TezRs, which could technically affect cell surface-bound primary TezRs. However, comparing the morphology of biofilms formed by *C. albicans* TezR–D1$^d$ with those of TezR–D2$^d$ and TezR–D1$^d$/D2$^d$, or comparing *C. albicans* TezR–R1$^d$ with TezR–R2$^d$ and TezR–R1$^d$/R2$^d$ (corresponding to the destruction of primary, secondary, and combined primary and secondary TezRs), differences could be clearly observed. From a technical standpoint, the treatment of fungi with nucleases (leading to the destruction of primary TezRs) resulted in different effects compared with the fungal growth on solid nutrient media supplemented with nucleases (leading to the inactivation of secondary TezRs). Taken together, these results indicate that each primary and secondary TezR has its own individual regulatory role that affects the activation of particular genes involved in fungal biofilm morphology.

### 3.7. Control of Viral–Host Interactions by TezRs

Our previous studies demonstrated that TezRs could control the interaction between bacteriophages and bacterial hosts [51]. To confirm that TezRs could also control the interaction between viruses and eukaryotic cells and further characterize this process, we determined that the destruction of TezRs could inhibit viral replication. Thus, Vero cells were infected with HSV-1 at a multiplicity of infection (MOI) of 0.1, and primary TezRs were destroyed 2 h post-infection (h.p.i) to allow virus binding to the cell surface and internalization (Figure 7A). In this model, Vero cells, following the individual loss of TezR–D1, were still subjected to observable cytopathic effects (CPEs). However, CPEs were inhibited after the loss of TezR–R1 or combined inactivation of TezR–D1/R1.

Subsequently, we studied how TezR destruction at 2 h.p.i. would affect the viral yield (Figure 7B). Surprisingly, the viral titers of HSV-1 were reduced only in Vero TezR–D1$^d$/R$^d$ cells ($p < 0.05$), whereas Vero TezR–R1$^d$ cells had the same viral yield in the supernatant as that in cells with unaltered TezRs. Together, these results confirm that different TezRs regulate viral–host interactions, including blocking certain post-entry stages of the HSV life cycle.

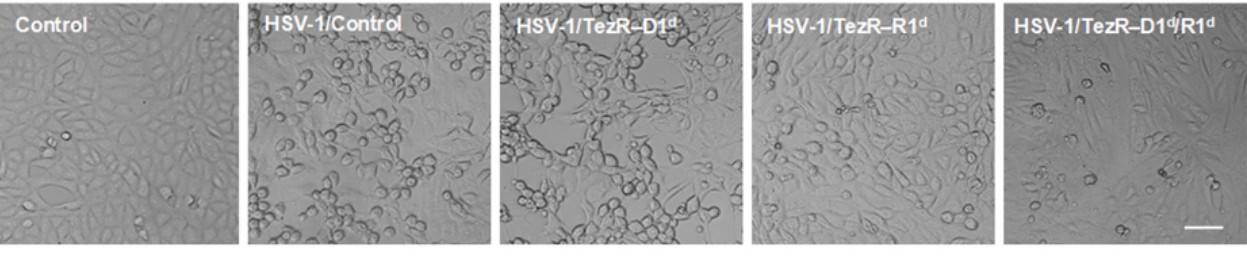

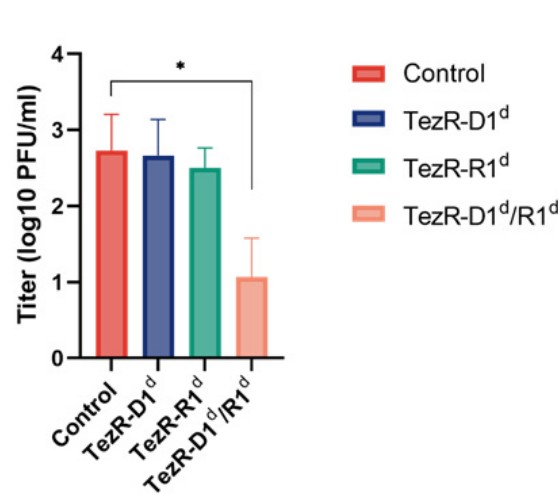

**Figure 7.** TezRs participate in virus–host interactions. Control Vero cells were infected with HSV-1 at an MOI of 0.1. They were either treated with water (control), or primary TezRs were destroyed 2 h post-infection (h.p.i.). (**A**) The morphological changes indicated a reduction in the cytopathic effect (CPE) in Vero cells following the destruction of TezRs captured at 48 h.p.i. (magnification, ×10). The scale bars represent 100 μm. (**B**) The virus in the supernatant was harvested at 48 h.p.i. and subjected to titration. Data are expressed as the mean ± SD ($n = 3$). * $p < 0.05$ as compared to the control Vero cells infected with HSV-1.

### 3.8. TezRs Regulate Cell Response to Insulin and Tramadol

Next, we examined the effect of TezR destruction on the cell response to insulin and tramadol. To examine the role of TezRs in regulating insulin sensitivity, we treated CHO cells with an insulin-transferrin-selenium (ITS) mix, for which insulin is known to have a primary growth-stimulatory effect [56,77]. A morphological analysis revealed that the treatment of water-treated control cells with unaltered TezRs with ITS in FDS-free medium increased cell adhesion, and the loss of primary DNA- or combined DNA- and RNA-formed TezRs did not affect the sensitivity of cells to ITS. Therefore, CHO TezR–D1$^d$ or TezR–D1$^d$/R1$^d$ cells exhibited the same pattern of cell adhesion in the presence of ITS. However, following the loss of TezR–R1, treating CHO cells with ITS increased the number of adherent cells by 331% ($p < 0.001$; Figure 8A,B). One explanation for the observed effect could be that if primary RNA-formed TezRs normally regulate insulin receptors, following their destruction, insulin receptors could have altered activity.

Importantly, the combined destruction of TezR–D1 and TezR–R1 did not affect the number of attached cells. This indicates that the increase in the pro-adhesive activity of ITS could occur only if, cells still had TezR–D1 along with TezR–R1 destruction. Taken together, these observations indicate that cell responses to insulin depend on the presence TezRs.

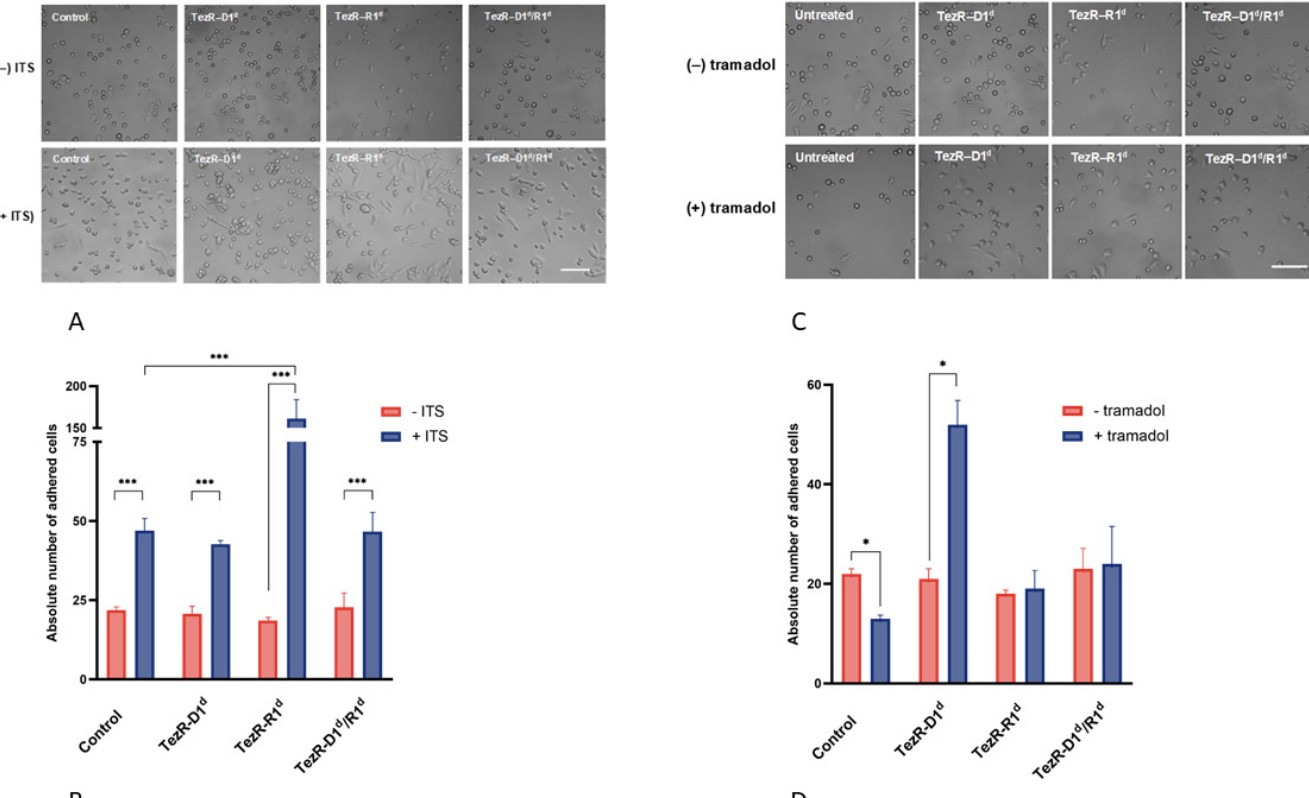

**Figure 8.** Effect of TezR destruction on cell responses to insulin and tramadol. (**A**,**B**) Effect of TezR destruction on the effects of insulin on CHO cells. (**A**) The images depict a visual difference in adherent cell numbers at 48 h following equal plating of CHO control, TezR–D1$^d$, TezR–R1$^d$, and TezR–D1$^d$/R1$^d$ cell (pictures acquired using a 10× objective). The scale bars represent 100 μm. (**B**) Quantitatively, total cell counts of adherent cells in a field view at 48 h increased for naïve cells and cells without TezRs in the presence of insulin-transferrin-selenium (ITS). The destruction of TezR–R1 also significantly increased attached cell numbers compared with that in control cells in the presence of ITS (mean ± SD from 20 fields of view from three independent experiments). *** $p < 0.001$. (**C**,**D**) Effect of TezR destruction on the response of T98G cells to tramadol. (**C**) T98G cells with unaltered or lacking primary TezRs were plated in media supplemented with 200 μM tramadol, and the efficacy of cell adhesion during the first 3 h was compared to that in cells plated in the same media but without of tramadol (magnification, ×10). The scale bars represent 100 μm. (**D**) The bars represent the total numbers of adherent cells in a field of view (mean+SD from 20 fields of view from three independent experiments), demonstrating that the destruction of TezR–R1 or TezR–D1/R1 prevented the inhibition of cell adhesion mediated by tramadol, and the inactivation of TezR–D1 increased growth in the presence of tramadol. * $p < 0.05$.

The role of TezRs in the regulation of cell sensitivity to opioids was studied by evaluating the responses of T98G cells to tramadol. In the present study, T98G glioblastoma cells expressing opioid receptors, following the destruction of primary TezRs, were plated on media supplemented with tramadol, and growth was compared to that of the same cells in media without tramadol [78–80]. Tramadol treatment showed an inhibitory effect on cell attachment with unaltered TezRs but not on that when TezRs were destroyed (Figure 8C,D). Thus, T98G TezR–R1$^d$ and T98G TezR–D1$^d$/R1$^d$ in the presence of tramadol showed the same growth pattern as that of cells without tramadol. Surprisingly, T98G TezR–D1$^d$ cells cultured in the presence of tramadol attached and flattened on the substratum more rapidly than when cultured in media without tramadol. These results suggest that cells without primary TezRs do not react to the tramadol-induced inhibition of cell adhesion, indicating that TezRs supervise the regulation of cell responses to tramadol.

### 3.9. TezRs Control the Response of Mammalian Cells to Visible Light

Given the previously discovered role of TezRs in microbial light sensing, we next evaluated their implications in visible light responses in mammalian cells [40]. We analyzed the differences in adhesion of Vero cells following TezR destruction when grown in the dark or in visible light for 24 h. We observed phototoxic effects on water-treated Vero cells with only a few cells adhering after growth in the light environment (Figure 9A,B). Meanwhile, the loss of TezR–D1 did not lead to any protection from light, and Vero TezR–D1d cells exhibited the same pattern of phototoxic effects in a light environment as that of vehicle-treated cells. However, the loss of TezR–R1 reduced the detrimental effects of light. Vero TezR–R1d cells attached and grew with a similar efficiency as that in the dark, although they were smaller in size and less flattened (Figure 9A–C). The destruction of TezR–D1/R1 protected cells from light-induced inhibition; however, this protection was less pronounced than that with the destruction of TezR–R1$^d$ (Figure 9A–C). Comparing the growth of Vero TezR–R1$^d$ and Vero TezR–D1$^d$/R1$^d$ cells under light, we observed that within 24 h of Vero TezR–R1$^d$ culture, the majority of cells had a triradiate morphotype, whereas all Vero TezR–D1$^d$/R1$^d$ had a fusiform shape. Together, these results suggest that TezRs–R1 is required for cells responses to light and plays an important role in photoprotection from light-induced cytotoxicity.

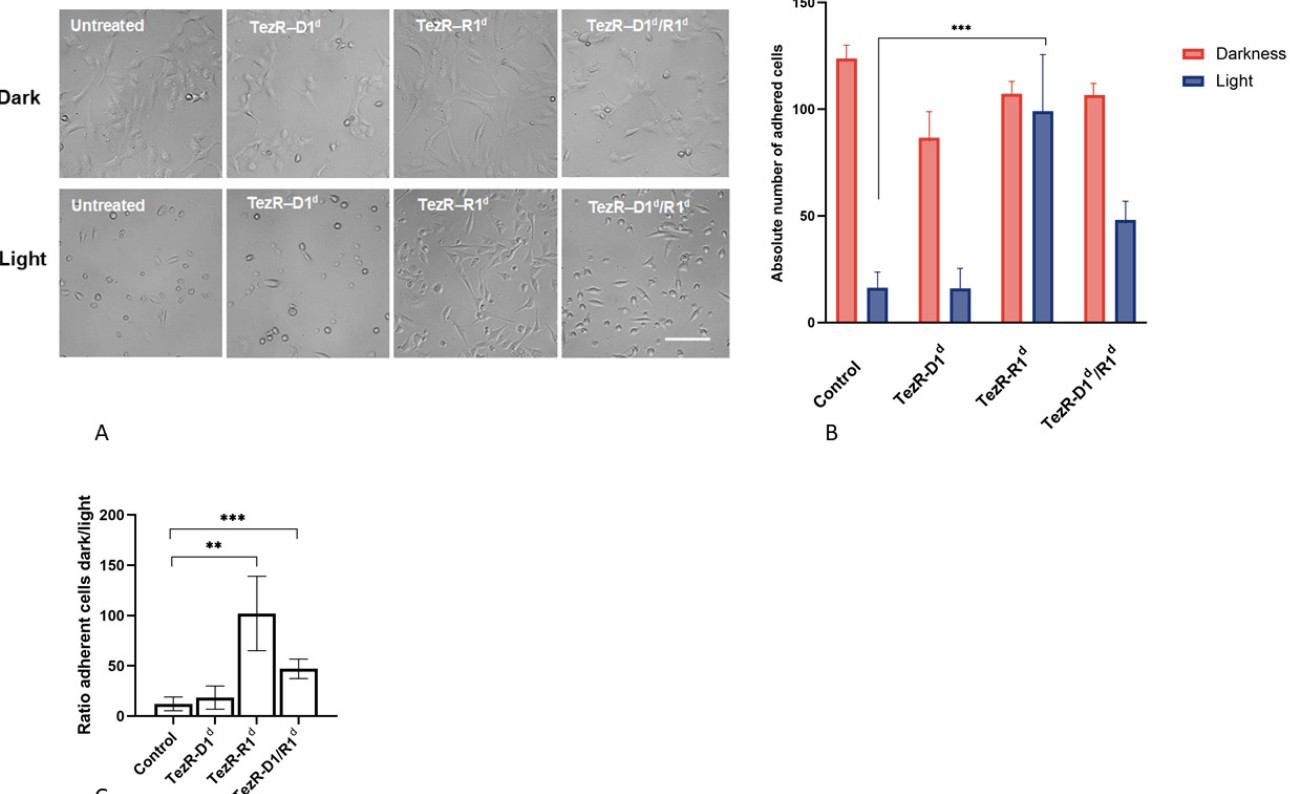

**Figure 9.** TezRs modulate light-induced cytotoxicity of mammalian cells. (**A**) Images of Vero control, TezR–D1$^d$, TezR–R1$^d$, and TezR–R1$^d$/D1$^d$ after 24 h incubation in darkness or under visible light. Magnification, ×10. The scale bars represent 100 μm. (**B**) The number of total adherent cells in a field view (mean ± SD from 20 fields of view from three independent experiments), showing that the destruction of TezR–R1 decreased light-induced phototoxicity. *** $p < 0.001$. (**C**) The ratio of adherent cells in the dark to those in the light for 24 h (mean ± SD from 20 fields of view from three independent experiments). ** $p < 0.01$ *** $p < 0.001$.

### 3.10. Role of TezRs in Eukaryotic Cell Temperature Tolerance

Next, we studied the implication of TezRs in eukaryotic cell tolerance to higher temperatures. Water-treated, control *C. albicans* exhibited maximum tolerance up to 55 °C, whereas

the destruction of primary DNA-formed TezRs alone or in combination with primary RNA-formed TezRs, with the formation of *C. albicans* TezR–D1$^d$ and TezR–D1$^d$/R1$^d$, impeded survival at temperatures greater than 53 °C (Figure 10A). The destruction of primary RNA formed by TezRs increased cell tolerance to heating. Thus, *C. albicans* TezR–R1$^d$ was able to survive at temperatures up to 58 °C. By analyzing the dynamics of fungal cell death at different temperatures, we found that control, water-treated *C. albicans* had a reduced CFU number by more than 4 log$_{10}$ units when heated to 50 °C and by 6 log$_{10}$ units when heated at 54 °C, whereas the CFU number *of C. albicans* TezR–R1$^d$ at 50 °C, and even at 54 °C was reduced by only 2 log$_{10}$ units.

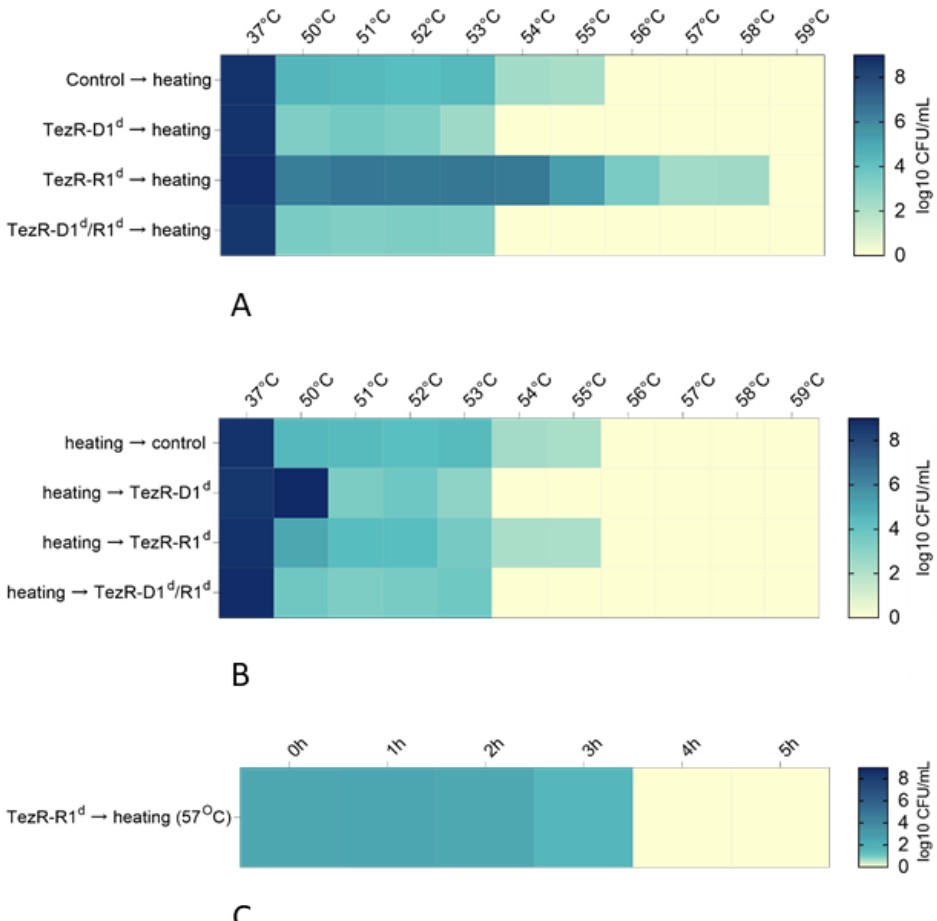

**Figure 10.** Role of TezRs in eukaryotic cell survival at higher temperatures. The heat map represents the effects of the destruction of different TezRs on the survival of *Candida albicans*. The vertical rows represent the temperatures of the heating and horizontal rows are the probes. The color intensity is an average log$_{10}$ CFU/mL number, represented by a color scale, from white (minimal) to blue (maximum). Data represent the average of three independent experiments. (**A**) TezRs were destroyed, and *C. albicans* was heated for 15 min. The control cells were treated with vehicle. (**B**) *C. albicans* was heated for 15 min, after which the TezRs were destroyed. Control cells were treated with vehicle. The destruction of TezRs did not affect the fungal sensitivity to heating. (**C**) Time of temperature sensitivity restoration in *C. albicans* following primary TezR destruction. Heatmap of time when the enhanced temperature tolerability of *C. albicans* disappeared following the destruction of TezR–R1. Each row is the hour at which each probe was heated after the destruction of TezR–R1 at the maximum temperature that was tolerated by *C. albicans* TezR–R1$^d$ (58 °C). The color intensity is an average log$_{10}$ CFU/mL number, represented by a color scale, from white (minimal) to blue (maximum). Data represent the average of three independent experiments.

Subsequently, we determined whether the observed, enhanced temperature survival was due to transcriptome-level responses triggered by TezR–R1 destruction or if they were attributed to the direct role of TezR–R1 in the regulation and sensing of temperature alterations. To that end, we heated untreated *C. albicans* at different temperatures and destroyed primary TezRs immediately after heating to alter fungal characteristics (i.e., transcriptional machinery). As shown in Figure 10B, the loss of primary TezRs after heating had no effect on the increase in temperature tolerance. Thus, we showed that the response of fungi to higher temperatures is regulated by primary RNA-formed TezRs; specifically, these responses are attributed to the presence or absence of TezRs at the time of heating and not to the sole transcriptome alterations initiated by their inactivation.

Then, we determined how much time is required for fungi that had become resistant to heating after primary TezR destruction to recover normal temperature-sensing capacity, and used this as a surrogate marker of the possibly time of functional restoration of active cell surface-bound TezRs. After the destruction of primary RNA-based TezR, *C. albicans* was inoculated into Sabouraud broth and allowed to grow with repeated sample collection hourly, each time being heated at the maximum temperature that was tolerated by *C. albicans* TezR–R1$^d$ (58 °C). Each hour after heating, *C. albicans* was inoculated into Sabouraud broth to identify the presence or absence of fungal growth after 24 h at 37 °C. The presence of fungal growth after heating at any time point indicated that *C. albicans* still had enhanced temperature survival, and this was taken as a surrogate marker of the absence of normal response restoration following TezR–R1 loss. In turn, the absence of fungal growth at certain time points suggested that the functionally active TezR–R1 was restored and fungi could normally sense and respond to the higher temperature. As demonstrated by this experiment, 4 h was required for *C. albicans* to restore functionally active TezR–R1 and normal temperature tolerance (Figure 10C).

*3.11. TezRs Control Sensitivity to UV*

To determine the role of primary TezRs in the sensitivity of fungal cells to UV, we investigated whether the inactivation of TezRs would affect *C. albicans* survival after exposure to UV from 60 to 100 s (Figure 11A). The maximum tolerable UV exposure for control fungi was 80 s, whereas the maximum tolerable duration of UV exposure for TezR–D1$^d$ and TezR–R1$^d$ cells was 70 and 90 s, respectively. Therefore, the removal of TezR–D1 increased cell sensitivity to UV, whereas the destruction of TezR–R1 decreased it. The destruction of both TezR–D1/R1 did not affect the maximum tolerable UV exposure.

To differentiate the effect of the presence of primary TezRs on response to UV radiation from alterations caused by the transcriptome-level responses triggered by TezR destruction, untreated *C. albicans* was exposed to different UV levels and TezRs were destroyed after that (Figure 11B). We observed that the loss of primary TezRs after UV irradiation had no effect on the increase in UV tolerance. Taken together, these results show that primary TezRs are implicated in the response of cells to UV radiation.

*3.12. TezRs Are Implicated in Magnetoreception in Eukaryotic Cells*

We also analyzed the role of TezRs in magnetoreception by analyzing its effects on *C. albicans* growth under shielded geomagnetic conditions. The inhibition of the geomagnetic field by cultivation in μ-metal for 6 h resulted in increased growth of control, water-treated *C. albicans* with an almost 4-fold increase in the OD600 value (Figure 12).

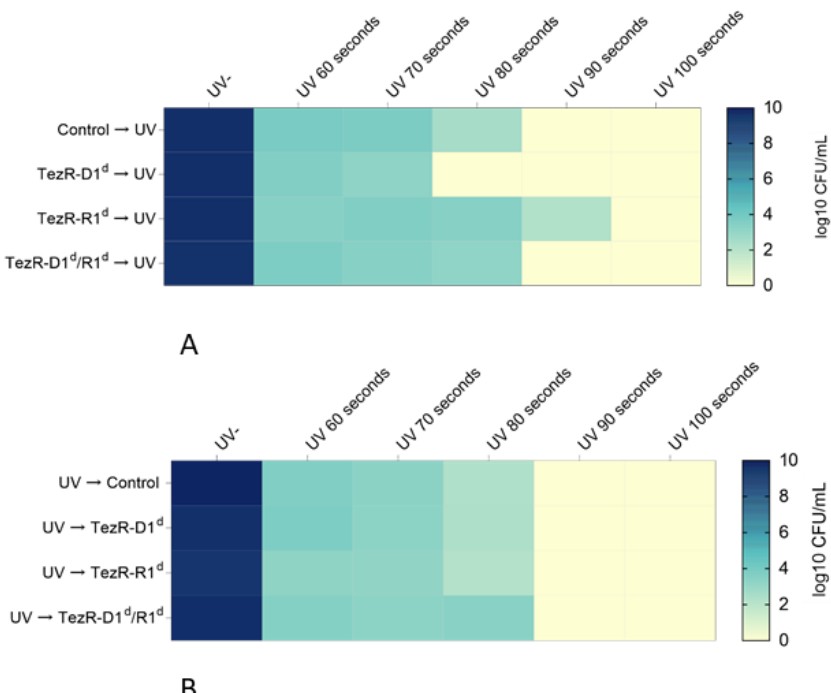

**Figure 11.** Role of primary TezRs in resistance of Candida albicans to UV exposure. The heat map represents the difference between live fungi, measured as viable counts (log$_{10}$ CFU/mL) before and after UV exposure. The vertical rows represent the duration of UV exposure, and the horizontal rows represent the probes. The color intensity is the average log$_{10}$ CFU/mL numbers, represented by a color scale, from white (minimal) to blue (maximum). (**A**) The destruction of TezRs was performed prior to UV irradiation. The destruction of TezR–R1 increased the tolerance of *C. albicans* to UV-induced death. (**B**) The destruction of TezRs prior to UV irradiation did not affect fungal sensitivity to UV irradiation. Data represent the average of three independent experiments.

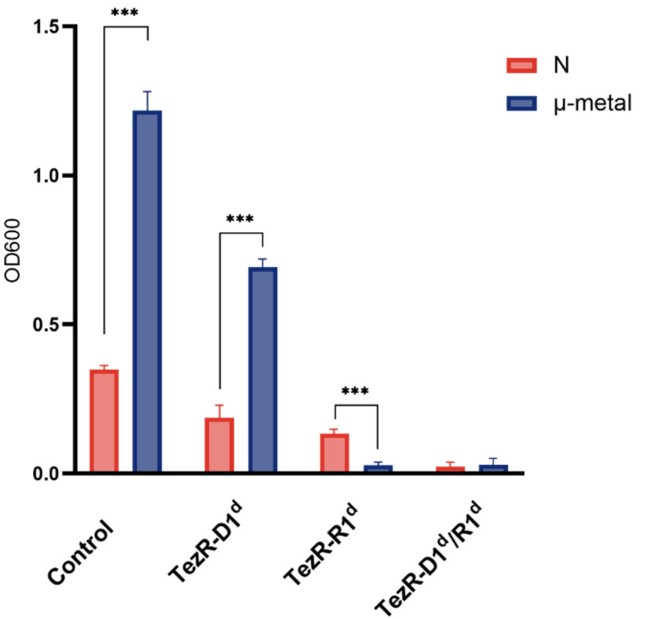

**Figure 12.** Effect of TezRs on magnetoreception in eukaryotic cells. The bars represent the effects of different TezRs destruction on planktonic growth under normal (N) and inhibited (μ-metall) geomagnetic conditions at 12 h growth (mean + SD from three independent experiments), *** $p < 0.001$.

Following the destruction of TezR–D1, *C. albicans* placed in the inhibited geomagnetic field also exhibited an increased OD600 value compared to that in cells cultivated under unaltered magnetic conditions. However, the cultivation of *C. albicans* in μ-metal following the loss of TezR–R1 resulted in the opposite effect. Thus, by the end of the observation period, *C. albicans* TezR–R1$^d$ had reduced biomass when grown in the μ-metal compared to that under normal geomagnetic conditions. Under the same conditions, the inhibition of the geomagnetic field did not increase the growth of *C. albicans* following the loss of both TezR–R1/D1. Altogether, these data suggest that primary RNA-formed TezRs are implicated in the normal response of cells to the geomagnetic field.

*3.13. TezRs Are Implicated in the Utilization of Nutrients*

Next, we studied the role of TezRs in nutrient utilization. *C. albicans* with inactivated primary TezRs was inoculated in M9 minimal media supplemented with maltose as the sole carbon source [81]. We compared the lag phase, which was taken as the period between inoculation of fungi and the initiation of biomass growth, and included the time required for sensing maltose and gene expression changes required to start its utilization [82,83]. We found that the loss of TezR–D1 did not affect the time lag of *C. albicans* (Figure 13A). In marked contrast, *C. albicans* with destroyed TezR–R1 alone or in combination with TezR–D1 (Figure 13B,C) displayed a longer time lag, indicating that these cells needed more time to start maltose utilization. Thus, *C. albicans* TezR–R1$^d$ displayed a delayed time lag of more than 3 h, whereas that of *C. albicans* TezR–D1$^d$/R1$^d$ was over 4 h, when compared to the time required for control fungi.

Subsequently, we confirmed that the observed increase in the time lag following the removal of TezR–R1 is associated with sensing and regulatory roles in nutrient utilization and that it is not a result of toxic effects and decreased cell activity triggered by TezR inactivation. We conducted experiments designed to prove that if *C. albicans* requires the presence of TezR–R1 to initiate maltose utilization, then fungi with unaltered TezR–R1 and pretreated with maltose followed by TezR–R1 loss will have a time lag similar to that of control *C. albicans* with unaltered TezRs. In other words, this experiment had to show that once fungi are engaged in maltose, they will continue to respond to it, even if TezR–R1 is subsequently destroyed (Figure 13D). In agreement with this hypothesis, control *C. albicans* exposed to maltose for 2 h with subsequent TezR–R1 destruction and the inoculation of resulting *C. albicans* TezR–R1$^d$ into M9 supplemented with maltose resulted in growth similar to that of the control *C. albicans*. In summary, these results support the critical role of TezRs in nutrient utilization.

*3.14. Implication of the TR-System in Cell Memory Formation and Forgetting*

Given the identical role of TezRs in eukaryotes and prokaryotes in sensing and utilization nutrients, we reasoned that in eukaryotes, TezRs might also play a role in cell memory formation. We used an 'adaptive' memory experiment to evaluate whether TezRs are implicated in remembering previous engagement with nutrients in eukaryotes, as we performed previously in bacteria [84].

We explored if TezRs are implicated in this memory of previous engagement with nutrients. For this goal we exposed "maltose-naïve" and "maltose-sentient" *C. albicans* with unaltered TezRs to maltose for different time periods. Subsequently, TezR–R1 was removed, and fungi were placed in fresh M9 media containing maltose (Figure 13E). We found that for maltose-naïve control *C. albicans*, the minimal time of first exposure before the destruction of TezR–R1 required to trigger maltose utilization was 2 h. The situation was dramatically changed in maltose-sentient *C. albicans* (Figure 13F), for which exposure to maltose for only 0.5 h was sufficient to sense and trigger maltose utilization, after which the destruction of TezR–R1 did not affect this process. The difference in time required for cells to possess an unaltered TezR–R1 at first (2 h) and repeated (0.5 h) contact with maltose, to sense and trigger its utilization, indicates the involvement of the TR-system

in long-term cell memory formation, enabling a faster response time and the capacity to adapt to behavior according to past experience [85].

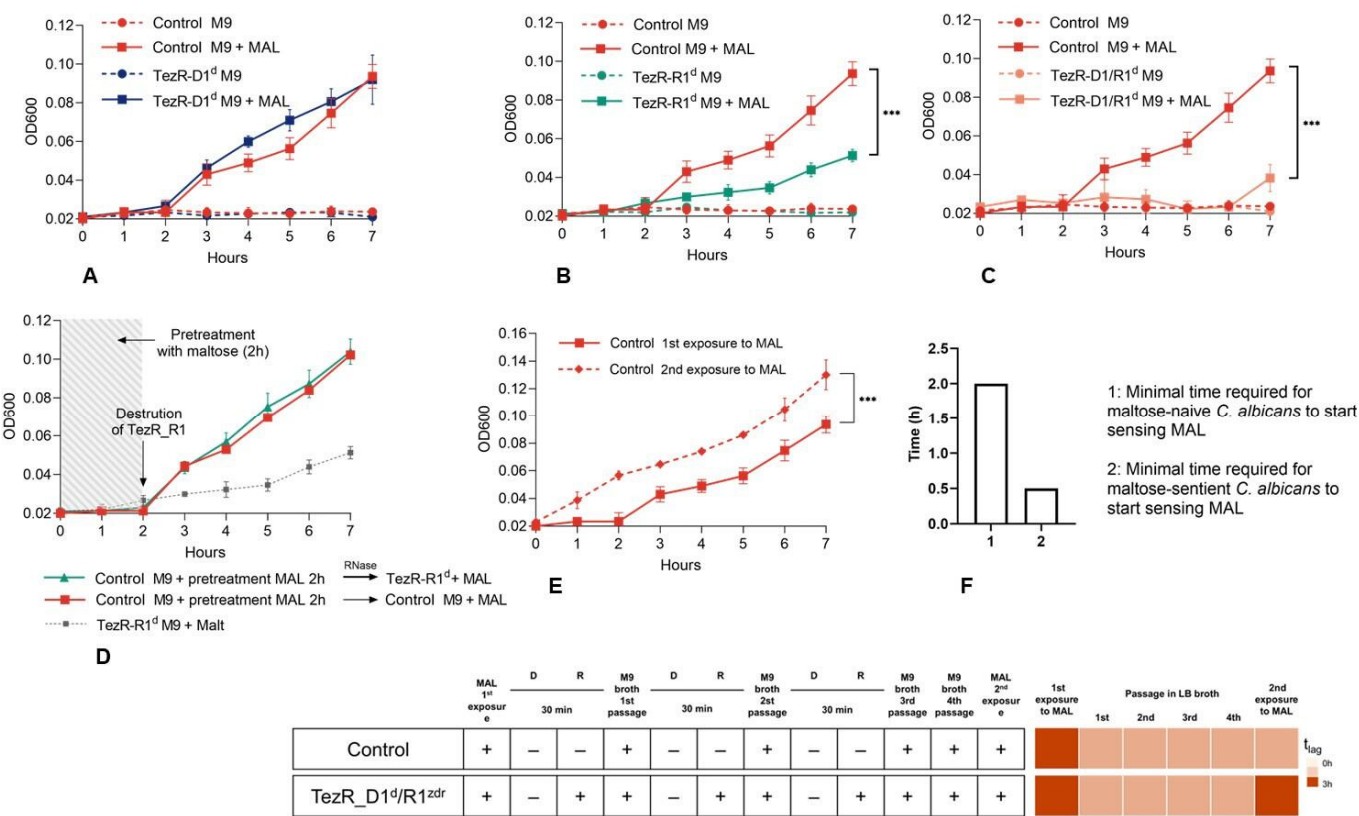

**Figure 13.** The role of TezRs in chemosensing and cell memory. Growth of control *Candida albicans* or cells following the removal of primary TezRs on M9 medium with and without maltose (MAL), as monitored over time. Supplementation with MAL was critical for supporting fungal growth on M9 minimal medium. The time from the beginning of maltose utilization when exponential growth began corresponded to the end of the time lag. (**A**) *C. albicans* TezR–D1$^d$ grown in M9 medium with or without MAL; (**B**) *C. albicans* TezR–R1$^d$ grown in M9 medium with or without MAL; (**C**) *C. albicans* TezR–D1$^d$/R1$^d$ grown in M9 medium with or without MAL. (**D**) Pretreatment of control *C. albicans* with MAL within 2 h followed by TezR–R1 destruction and subsequent growth on M9 media supplemented with MAL; (**E**) The difference in time required for control maltose-naïve *C. albicans* and control maltose-sentient *C. albicans* to start growing on M9 media with MAL. The lag phase, taken as the time to start dexamethasone utilization, was 2 h shorter in maltose-sentient *C. albicans*. (**F**) Minimal time required for *C. albicans* to start utilization MAL. The *Y*-axis represents the time lag of control *C. albicans* upon initial and second exposure to MAL. (**G**) The role of TezRs in cell memory and forgetting based on the time of initial MAL utilization (time lag) for maltose-naïve and maltose-sentient *C. albicans*. The remaining part of the table shows the protocol. The first exposure of control *C. albicans* to MAL in M9 minimal media (time lag, 3 h) led to propagation across multiple generations resulting in faster rates of population growth after the second exposure to MAL in M9 broth with a time lag of 1 h. Two repeated rounds of TezR–R1/D1 inactivation between the first and second exposures to MAL, followed by growth in nutrient medium without nuclease, did not affect cell memory to maltose. However, three cycles of TezR–D1/R1 removal resulted in a loss of memory to previous MAL exposures. Thus, after three cycles of TezR–D1/R1 loss, *C. albicans* behaved like cells that had not been expose to MAL with a time lag of 3 h. The time lag after each passage in M9 media with or without MAL is represented as a heatmap on the right part of the image based on a color scale, from white (0 h) to red (3 h).

Similar to that in bacteria, we next confirmed the role of the TR-system in forgetting. We conducted multiple rounds of combined TezR destruction followed by a wash-out period in growth medium without maltose. As depicted in Figure 13G, one or two cycles of TezR destruction and restoration for 24 h did not affect the memory of maltose-sentient fungi, and the time lag of such cells during the second maltose exposure was shortened, compared with that in maltose-naïve cells. However, three rounds of destruction of TezR–D1/R1, followed by their restoration, thus putting the cells in to a so-called "zero" state led to the forgetting of the previous exposure to maltose. Thus, the behavior of *C. albicans* TezR–D1$^{zdr}$/R1$^{zdr}$ at the second contact with maltose was similar to that of control *C. albicans* at the first maltose exposure, with a time lag of 3 h.

### 3.15. Role of Transcriptase Inhibitors in the TR-System of Eukaryotic Cell

We hypothesized that the formation and function of TezRs could be associated with reverse transcription. We thus studied how reverse transcriptase inhibitors (RTIs) affect cells following the destruction of different TezRs in *C. albicans*. We used representative of nucleoside analog inhibitors (NRTIs) and non-nucleoside inhibitors (NNRTIs) because recent data have confirmed that reverse transcriptases share similarity across different organisms and can be inhibited by some RTIs originally developed against human immunodeficiency virus (HIV) RT [86–89].

In this experiment, we used low doses of tenofovir, lamivudine, and etravirine, which were at least a 100-fold less than their MICs and did not show any inhibitory effects against *C. albicans* (the MICs of these RTIs were over 500 µg/mL as the highest concentration tested) (Supplementary Table S2). We analyzed their effects on fungal growth during the first 7 and 12 h, since this time is critical for cells to adapt to the new environment with critical reliance on TezRs. As shown in Figure 14A, although all three RTIs had an inhibitory effect during the first 7 h of *C. albicans* growth, by 12 h they did not show any growth inhibitory effects, and even some increase in fungal biomass was noted.

Studying the effect of RTIs on *C. albicans* with inactivated TezRs, we found that they had a statistically significant effect on cell growth with destroyed TezR–R1 (Figure 14A). In these settings, etravirine, tenofovir, and lamivudine significantly inhibited the growth of *C. albicans* TezR–R1$^d$ after 12 h of growth (all $p < 0.05$; Supplementary Table S3). Given that fungal growth is dependent on the presence of TezRs, the observation that RTIs had different effects on fungi with unaltered TezRs and on fungi following TezR–R1 inactivation suggest the function of RT in the TR-system.

### 3.16. Effects of Integrase Inhibitors on Signal Realization in the TR-System

Subsequently, we hypothesized that the TezR signaling pathway (cascade) includes the functions of recombinases and thus can be affected by integrase inhibitors. We used raltegravir, which is an inhibitor of HIV integrase and also known to interact with different recombinases, even with a distant structure from that of HIV integrase [90,91]. We used raltegravir at nontoxic concentrations (Supplementary Table S2) of less than 100 times its MIC. As previously shown in the section "Implication of the TR-system in cell memory formation and forgetting", maltose-naïve cells with unaltered TezR–R1 required 2 h to trigger maltose utilization. We assumed that the possible inhibitory effects of raltegravir on signal realization from TezR–R1 inside the cells might occur within this time period.

Thereafter, we studied how different periods of raltegravir addition from 0 to 120 min could affect maltose utilization. We found that raltegravir inhibited *C. albicans* growth on M9 with maltose when added together or within the first time periods but stopped having an inhibiting role once it was added starting from 120 min of growth (Figure 14B). These results indicate that recombination processes within the TR-system might be implicated in the realization of signals from TezRs.

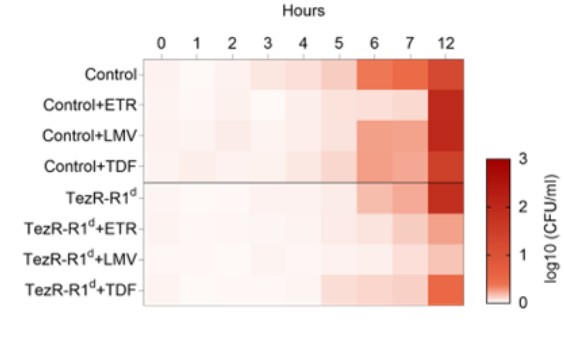

A

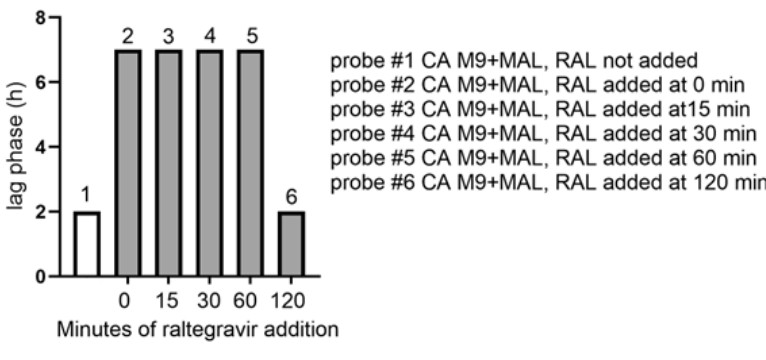

B

**Figure 14.** The effects of reverse transcriptase and integrase inhibitors on cells. (**A**) The effects of RT inhibitors on fungal growth depends on the state of TezRs. The heatmap representation of growth upon treatment with RT inhibitors in control *C. albicans*, *C. albicans* TezR–D1$^d$, *C. albicans* TezR–R1$^d$, and *C. albicans* TezR–D1$^d$/R1$^d$ grown in broth. Etravirine (ETR), lamivudine (LMV), and tenofovir (TDF) were added to the broth, and fungal growth at OD600 was monitored for 12 h and recorded at different time intervals. Vertical rows indicate hours and horizontal rows indicate probes. The OD600 was labeled based on a color scale, from white (minimal) to red (maximum) Data are representative of three independent experiments. (**B**) Raltegravir (RAL) added together, 15, 30, or 60 min after *C. albicans* (CA) growth on M9 media, with maltose (MAL) increased the lag phase, taken as the time required for fungi to initiate maltose utilization. When RAL was added 120 min after the plating of control *C. albicans* on M9 with MAL (black triangles with red line), it did not affect the start of MAL utilization.

### 3.17. TezRs Regulate Multicellular Eukaryotic Structures

Subsequently, we studied whether TezRs are involved in the regulation of multicellular structures using wheat (*Triticum aestivum*) seeds as a study object. The shoot and root lengths of *T. aestivum* were significantly increased on the 7th day of cultivation after incubating the seeds with DNase for 30 min and the inactivation of DNA-based TezRs (Figure 15A,B). The average stem and root lengths were 24.9 cm and 16.3 cm, respectively, for seedlings from TezR–D1$^d$ seeds, which represented an improvement of approximately 57.9% and 27.5%, as compared to those for for water control samples ($p < 0.05$). The stem and root lengths were not altered compared to those in the control seedlings obtained from seeds with destroyed TezR–R1 or TezR–D1/R1.

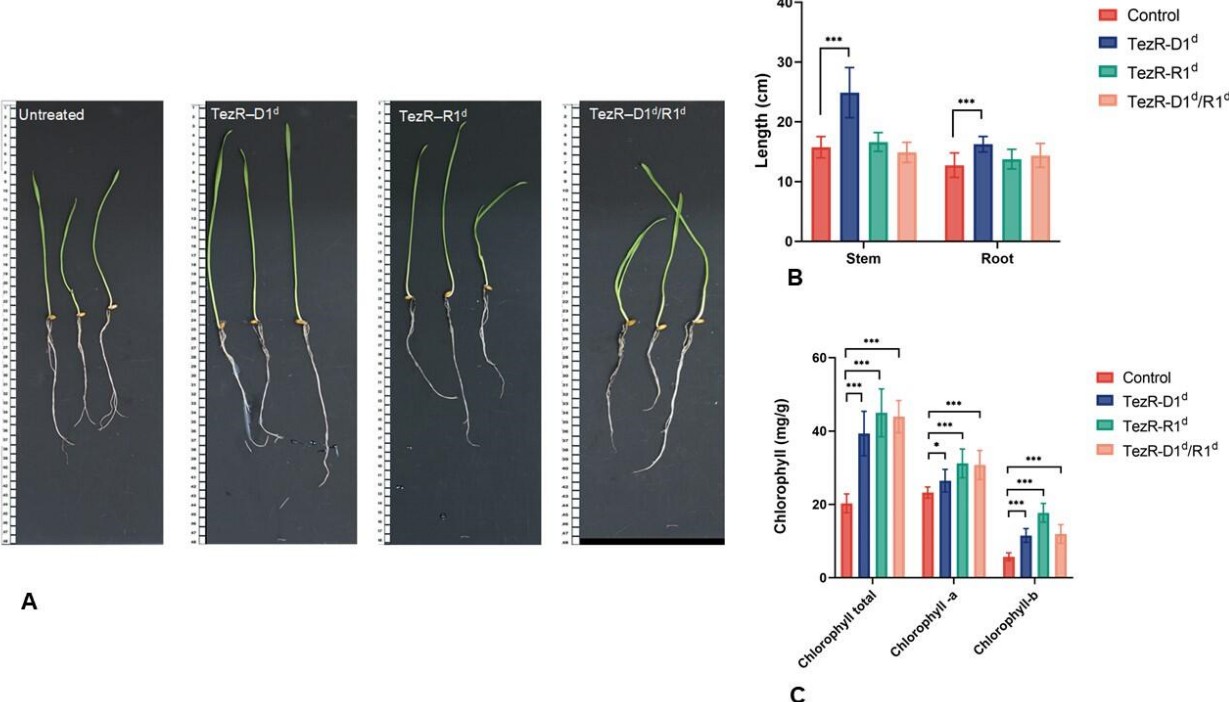

**Figure 15.** The effect of TezR inactivation in seeds of *Triticum aestivum* on cultivation. (**A**) Morphological appearance and growth differences of *Triticum aestivum* seedlings after DNA, RNA- or DNA- and RNA-based TezR inactivation, measured on day 7. The bar on the right shows the length in cm. (**B**) The average stem and root length (cm) on day 7 as the mean $\pm$ SD. (**C**) Chlorophyll a, b, and total chlorophyll (mg/g) levels on day 7. Data represent results from three independent experiments as the mean $\pm$ SD. * $p < 0.05$, *** $p < 0.001$.

The chlorophyll levels in *T. aestivum* seedlings obtained from seeds with inactivated TezRs are shown in Figure 15C. The destruction of any TezRs increased chlorophyll a, b, and total chlorophyll content in seedlings compared to that in the control group. It was not the goal of this study to evaluate the specifics of nuclease penetration inside seeds and the destruction of TezRs, based on which specific structures potentiated seedling growth [92]. Data received for the first time indicate that some TezRs are implicated in the regulation of multicellular structures, enabling the regulation of not only single cells but also multicellular plant embryos.

## 4. Discussion

Here, we describe for the first time a newly discovered global TR-system in uni- and multicellular eukaryotes of different kingdoms that orchestrates major vital functions and responses and shares similarity with the identical system that we have recently identified in procaryotes [51]. This system is formed by nucleic acid-based receptors named TezRs and includes the work of reverse transcriptases and integrases. We classified TezRs based on the features of the DNA- and RNA-formed domains located outside the membrane as primary and secondary, highlighting a novel function of nucleic acids as receptors not related to protein synthesis. Using fungi, mammalian cells, and plants as model organisms, we have shown that TezRs are implicated in reactions to different chemical, physical, and biological factors, and their destruction leads to altered biological and synthetic activities, including massive transcriptome modifications.

We observed individual responses to TezR loss in different mammalian cell lines. Specifically, the removal of TezRs from Hek293T cells triggered apoptosis but did not affect the viability of Vero cells. Simultaneously, TezRs were shown to control certain steps of the cell cycle in Vero cells. The loss of TezR–D1 accelerated S phase progression in Vero cells,

which is responsible for the synthesis or replication of DNA. While on the one hand, much is known about the control of the G1/S and G2/M cycle, on the other hand, the mechanism of S/G2 transition control remains elusive, and thus, the implication of TezRs in this process is particularly interesting [93]. Based on the current knowledge of this process, we can only speculate that DNA-based TezRs might control S-phase length and modulate DNA replication by affecting the p53–DREAM pathway and S-phase checkpoint kinases [93–95]. On a transcriptome level, we observed alterations in *MN1* and *CCL2* gene expression that are known to participate in cell-cycle regulation; however, these are more known to be implicated in G1/S phase transition, and thus, their participation in the observed alteration remains uncertain [96,97].

The loss of any TezR in Vero cells increased cell proliferation and accelerated the migration kinetics in the gap filling assay. Since both of these events are essential for physiological processes, such as wound healing to re-establish tissue layers after injury and pathological processes that play a role in oncogenesis, these data indicate that TR-system could be implicated in their regulation [98–100]. Moreover, we obtained very similar results when TezRs were inactivated by binding with PI and not destroyed with nucleases. Therefore, we can assume that TezRs govern other known protein-based receptors that control cell migration found in Vero cells [101,102].

Importantly, in the current study, we treated cells with nucleases for a short period, which allowed us to study the responses of cells after a one-time TezR elimination step and to ensure that RNase is not internalized by cells. Studying the role of the TR-system in cell growth using fungi, we showed that it plays a previously unexplored and critical role in the regulation of adaptation to the outer environment including growth in liquid and solid media. The removal of different TezRs changed the dynamics of fungal growth in the lag phase when fungi adapt themselves to growth conditions. A similar pattern was observed when TezRs were studied in bacteria. In later stages of fungal growth, primary and secondary TezRs orchestrated budding and pseudohyphal and hyphal formation associated with changes in the cell cycle, which are known to be tightly regulated by hypha-specific gene expression [103,104]. TezRs also regulate fungal biofilm architecture, and TezR–R2 specifically controlled biofilm dispersal, which is an important element in fungal-caused pathogenesis [105,106]. Notably, we observed the same trend for the role of TezR–R2 in controlling bacterial biofilm dispersal.

Moreover, fungal biochemical activity is also controlled by the TR-system, alterations of which are specific following the destruction of different TezRs. We observed the activation of different hydrolytic enzymes following the loss of certain TezRs. Some activated enzymes are associated with the colonization and invasion of *C. albicans* into host tissues, whereas others are implicated in the nutrient assimilation process [107]. One of the most surprising results was that the destruction of TezRs could trigger the activation of genes required for the assimilation of urease and lactose by *C. albicans*, which are known not to be utilized by these fungi [73,75,108]. We reasoned that this could be realized only due to significant genomic rearrangements; however, a more detailed analysis will be performed in the follow-up research [109,110].

In our experiments, following combined treatment with DNase and RNase or treatment with PI, we generated cells in which both primary DNA- and RNA-based TezRs were inactivated. As in procaryotes, considering the broad diversity of factors controlled by TezRs, we assumed that these cells with inactivated primary DNA- and RNA-formed TezRs do not receive the necessary stimuli from the outer environment through TezRs, which might reveal different behavioral patterns, each of which would be paradoxical. Indeed, we observed that fungi lacking TezR–R1/D1 could not grow during the first hours and exhibited the significant depletion of synthetic activity, whereas mammalian cells started very early migration. In other cases, the behavior of cells following the destruction of TezR–D1/R1 had some features similar to the behavior of cells lacking either DNA- or RNA-based TezR, as well other individual patterns or even those of control cells. Thus, as for bacteria without TezR–D1/R1, we named eukaryotic TezR–D1$^d$/R1$^d$ cells as "Drunk

cells". In trying to address the reasons for such unique behaviors of "drunk cells", it is interesting to speculate that the inactivation of all primary TezRs could lead to the activation of some proposed internal (i.e., cytoplasmic) TezRs (TezR–i).

It has not escaped our attention that in eukaryotes like in prokaryotes, the TR-system was found to control the response of cells to different physical environmental stimuli, and the strong regulators that orchestrate these interactions were RNA-based TezRs. Since many papers show that under certain conditions, RNase can be internalized in mammalian cells, we ensured that under the conditions used in our experiments, RNAse activity was present only extracellularly [111]. We evaluated the role of the TR-system in the control of cell survival at high temperatures and found that fungi lacking TezR–R1 were less sensitive to heat-induced death at all temperatures studied, and moreover, they could survive at temperatures more than 3 °C higher than the maximum tolerable temperature of control cells. We reasoned that since the destruction of TezRs before heating increased cell survival, but after heating did not affect this process, this might implicate TezR–R1 in thermosensing and its effects on heat shock-regulated apoptosis and cell death. The idea that these newly discovered RNA-based TezRs can react to temperature is also supported by the notion that intracellular RNA could respond to an altered temperature through the modulation of translation [112–115].

To further confirm our findings regarding the universal role of TezR–R1 in the interaction with physical factors, we found that TezR–R1 also participates in cell responses to and survival upon UV exposure. The increased survival after UV irradiation in cells without RNA-based TezRs can be explained by the altered sensation of UV or modulation of cell responses to UV-induced damage, which largely depends on the repair of UV-induced DNA damage [116–118]. The destruction of TezRs modulated cell viability only when these TezRs were destroyed before and not after UV exposure, which adds another line of evidence specifically implicating TezRs in cellular responses to UV. UV reception is poorly studied, and currently, the only described ultraviolet photoreceptor, UVR8, has been found in plants [119]. Interestingly, some researchers have tried to characterize UV sensitivity through visible light receptors to determine if there is a photoreceptor that absorbs light in the UV region [120]. This was replicated in our study, as the destruction of TezR–R1 in Vero cells protected them from light-induced cytotoxicity and cells without TezR–R1 were able to effectively attach and grow, unlike that in controls, suggesting the role of TezR–R1 in sensing light [31,32].

Continuing to study the role of TezRs in eukaryotic responses to other physical factors, we found that the destruction of TezR–R1 resulted in an altered response of fungi to the inhibited geomagnetic field. While shielding the geomagnetic field in *C. albicans* with unaltered TezRs potentiated fungal growth, in marked contrast, cells lacking TezR–R1 did not react to this altered geomagnetic field. These results suggest that the primary TR-system might be involved in sensing and primary responses to the geomagnetic field. To date, the identity of a magnetic sensor in eukaryotic cells remains unclear; however, different mammalian and plant cells have been shown to sense geomagnetic fields [23,121,122]. Interestingly, the ability of TezRs to interact with a magnetic field could be explained by the nucleic-acid structure of these receptors, since some publications describe the paramagnetic properties of nucleic acids and their ability to emit or transmit waves [123–125].

We observed that the TR-system prevails in the response of cells to external stimuli, governing the functions of well-known receptor signaling pathways and modulating cell responses to hormones, which is particularly interesting in terms of the role of TezRs in multicellular organisms. Thus, in our study, we observed that the loss of TezR–R1 potentiated insulin-induced growth-promoting effects, which in experimental settings are known to be realized through a cascade of phosphorylation events leading to the activation of enzymes that control the uptake of uridine and glucose, ultimately enhancing the synthetic process and accelerating cell cycle progression [126–129]. Studying the role of the TezR in cell responses following the stimulation of μ-opioid receptors with tramadol, we found that the destruction of TezR–D1 in these settings decreased tramadol-induced inhibition of growth

meaning that the TR-system oversees certain μ-opioid receptor signaling pathways as well [78]. In addition, TezR–D1 determines mammalian cell interactions with biological factors such as viruses, controlling certain post-entry stages of the viral life cycle. This means that the TR-system controls viral transmission and can be viewed as an element implicated in the regulation of gene transfer across species and genetic information metabolism [130].

In the aforementioned studies, we found interesting phenomena that have not yet been completely evaluated. Some functions of both DNA- and RNA-based TezRs found in this work, including the regulation of cell cycle, cell migration, activation of μ-receptors, and response to insulin, are realized through the PI3K/Akt signaling pathway [78,129]. Dysregulation of PI3K/Akt signaling is known to be implicated in the transition of normal cells into a malignant state, leading to questions about the role of the TR-system in this process [15,99].

The sensory and regulatory roles of TezRs with respect to chemical factors were evaluated in fungi using maltose as the only growth substrate. The utilization of maltose was controlled by TezR–R1, as the loss of this receptor before contact with maltose inhibited substrate assimilation, whereas its removal after a certain time of contact did not affect this process. It is particularly intriguing that the TR-system was implicated in cell memory, since despite recent interest in this process and the great amount of work dedicated to this topic, there is still no complex understanding of how cell memory is formed, maintained, and erased [131–133].

The role of the TR-system in memory formation in response to previous events was proven by the possibility that this memory could be erased via continuous rounds of TezR destruction. In our hands, after one round of TezR destruction and subsequent restoration, the substrate-sentient cells retained memory and continued to react to the substrate faster than substrate-naïve cells. However, after three repeated rounds of TezR destruction, the cell forgot its initial contact with the substrate and required the same amount of time as substrate-naïve cells to initiate substrate utilization. We named these cells with memory erased by multiple rounds of TezR destruction "Zero cells". Therefore, we discovered, for the first time, that the TR-system controls memory to preceding events and forgetting, which could lead to future studies on cellular programming, which has recently attracted much attention [47,132]. Notably, a similar role of TezRs in memory formation and forgetting was observed in prokaryotes [51]. Such an adaptive memory of cells to a nutrient factor is similar to the adaptive strategy of B-cells, for which the secondary and more pronounced response is based on their affinity for antigens and the higher number of cells possessing relevant receptors [134–136].

To gain insight into how the signal from TezRs is transferred, we found that integrase inhibitors could block this process. In our studies, raltegravir blocked the cell responses that were found to be controlled by TezRs such as maltose utilization. These results point out that recombinases might be implicated in the processing of signals from TezRs inside the cells; however, the detailed mechanism remains enigmatic.

In this study, we also evaluated whether TezRs regulate the growth of multicellular eukaryotic organisms using a model of *T. aestivum* seeds, since as for now only a few receptors on seeds that regulate plant growth are known [137–139]. These results indicated that TezRs control not only individual cells but also multicellular structures.

Finally, to explain the pathways underlying the signaling origin and cascades from TezRs, we hypothesized that some of them might include RNA-dependent DNA polymerase and RNA-dependent RNA polymerase. Recent studies have shown the existence of multiple polymerases of these types in eukaryotic cells, including those with unknown functions [86].

In support of this idea, we used NRTIs and NNRTIs, which were originally developed to interfere with HIV-1 RT and were recently shown to have a broad spectrum of inhibitory activity for different non-viral polymerases [140–142]. We observed that RTIs differentially affected control cells with unaltered TezRs and cells following TezR–R1 loss. These data indicate that the recovery of cells following TezR destruction depends on RTs, but in this

study, we have not yet evaluated the molecular mechanisms underlying these effects. Interestingly, recent publications have indicated that some RTIs that inhibit endogenous RT activity modulate cell proliferation and differentiation, thus possessing anticancer effects, but the mechanism of such effects could not be clearly explained [89,143–146]. We speculate that at least some of these effects might be realized by the TR-system.

In our work on TezR destruction, we broadly used nucleases, thus indicative of real-life scenarios, since nucleases are broadly produced by different organisms. One can assume that nucleases of multicellular mammals might have previously overlooked roles by destroying the TezRs of certain cells [147,148]. In addition, the nucleotide composition of TezRs indicates that they most likely need to be encoded by the genome, and it seems logical that they would be localized in non-coding regions.

The existence of extracellular nucleic acids under normal conditions has been described in mammals, and their alterations are associated with different pathologies [102,148–153]. In particular, many of these alterations are associated with tumor progression [154–157]. The results of studies on the biological effects following the destruction of these extracellular nucleic acids indicate that due to the use of nucleases, in some of these experiments, TezRs could also be affected, but the effects of TezR loss could not be differentiated from the general effects of extracellular nucleic acid destruction.

Given the nucleic-acid based structure of TezRs, one can assume that they are evolutionary old, since they regulate protein-based regulatory pathways. This is also supported by the notion that according to the RNA world hypothesis, life on Earth began with an RNA molecule, and it can be assumed that this ancient RNA, along with the ability to copy itself without help from other molecules, could have had receptive functions [158].

We now face the need to develop new tools to study the TR-system. We believe that additional studies might reveal the currently overlooked specifics of their structure and localization, leading to the additional sub-classification of TezRs. For example, considering the global functions that are regulated by TezRs, we expect that certain differences in their biological activity can be achieved based on the physical characteristics of TezRs, such as their length, formation of loops, or alternative confirmations and structures such as DNA–RNA hybrids or triplexes [159–161].

How primary and secondary TezRs interact with cells and each other also remains an open question. We speculate that secondary TezRs might be free receptors that are not directly bound to the cell structures. Since secondary TezRs are eliminated by centrifugation, we believe they are localized away from cells, forming a type of a "receptive shield" around cells, and might interact with secondary TezRs of other cells, forming the most distant receptive network interacting with the outer environment.

We have not studied TezRs inside the cells, although we can expect the presence of TezRs on mitochondria and plastids as organoids that arose from the engulfment and integration of bacteria [162]. In addition, with the methods used in this study, we could not answer some questions that need to be addressed in follow-up studies, such as how TezRs interact with different chemical and physical factors and the specific signal trafficking through TezRs.

Future studies should unveil in more detail, the molecular mechanisms underlying the origin and the pathway for the translocation of TezRs to the cell surface, their specific effects on different cells including immune and malignant cells in addition to expanding our understanding of the entire set of sensing and regulatory processes in which they are involved. The interaction between TezRs with known protein-based receptors performing the same function is also a critical question that might pave the way to finding a more accurate way to control cell signaling and reception. Given that primary TezRs are associated with the cell membrane, they might also be involved in the functioning of ion channels; meaning that they can be used to better regulate cell membrane potential and play a role in neural communication and muscle contraction. Finally, the role of TezRs in the response of multicellular organisms is also intriguing.

Taken together, our data provide the first evidence of the existence of a novel receptive regulatory TR-system that participates in the recognition of environmental factors, the memory of them, and their forgetting. Considering the structural and functional similarity between prokaryotes and eukaryotes, the TR-system can be viewed as a global system with previously unknown pathways of interkingdom interactions. We believe that the existence of this type of global receptive system that modulates living objects will lead to the re-analysis of some previous studies from a new perspective.

**Supplementary Materials:** The following supporting information can be downloaded at: https://www.mdpi.com/article/10.3390/receptors1010003/s1, Supplementary Figure S1. The removal of cell surface-bound DNA and RNA molecules with nucleases. (A). Red fluorescence denotes cell surface-bound DNA and RNA of CHO cells stained with propidium iodide. (B). Green fluorescence denotes cell surface-bound DNA and RNA of Vero cells stained with membrane permeable SYTO. Supplementary Figure S2. Absence of RNase A internalization in Vero cells. Vero cells were treated with FITC labeled RNase A (50 μg/mL) for 30 min, after which RNase I was washed out and fluorescence was measured (A) immediately, or (B) after 24 h of growth. Supplementary Figure S3. The effect of primary TezR destruction on cell number and multipolar morphotypes vs. bipolar morphotypes in 24 h-old Vero cell cultures. The morphology of 24 h-old TezR–D1d, Vero TezR–R1d, and TezR–D1d/R1d Vero cells following treatment with nucleases, as discussed in the Materials and Methods, was studied by light microscopy. Cells were treated with vehicle as a negative control. Similar cellular morphology was observed in five independent experiments. Magnification: 10. Supplementary Figure S4. Absence of RNase A internalization in Candida albicans. C. albicans was (A) treated with FITC conjugated RNase A (50 μg/mL) for 1 h and fluorescence was measured immediately, (B) treated with FITC conjugated RNase A (50 μg/mL) for 1 h and fluorescence was measured after 6 h of growth, (C) cultivated on agar for 24 h, supplemented with FITC conjugated RNase A (50 μg/mL). Supplementary Table S1. The effect of primary TezR destruction on Candida albicans size. Supplementary Table S2. MICs of tested reverse transcriptase inhibitors and integrase inhibitors for control Candida albicans. Supplementary Table S3. The effect of reverse transcriptase inhibitors (RTIs) on control and Candida albicans TezR–R1d at 12 h of growth.

**Author Contributions:** V.T. and G.T. designed the experiments. V.T. and G.T. supervised the data analysis, analyzed the data and wrote the manuscript. All authors have read and agreed to the published version of the manuscript.

**Funding:** This research received no external funding.

**Institutional Review Board Statement:** Not applicable.

**Informed Consent Statement:** Not applicable.

**Data Availability Statement:** All the metagenomic datasets generated in this study are available upon request.

**Acknowledgments:** We gratefully acknowledge You Zhou, Microscopy facility at the Center for Biotechnology in University of Nebraska-Lincoln for help in microscopy; Kristina Kardava, Marya Vecherkovskaya for setting some experiments, as well as Tatiana Lazareva; Genome Technology Center (GTC) for expert library preparation and sequencing, and the Applied Bioinformatics Laboratories (ABL) for providing bioinformatics support and helping with the analysis and interpretation of the data. GTC and ABL are shared resources partially supported by the Cancer Center Support Grant P30CA016087 at the Laura and Isaac Perlmutter Cancer Center. This work has used computing resources at the NYU School of Medicine High Performance Computing (HPC) Facility.

**Conflicts of Interest:** The authors declare no conflict of interest.

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
