# Peer review of "Novel Cell Receptor System of Eukaryotes Formed by Previously Unknown Nucleic Acid-Based Receptors"

_2813-2564, doi:10.3390/receptors1010003_

Round 1
Reviewer 1 Report
The authors investigated the various functions of extracellular nucleic acid-based receptors. This paper is well experimented, and this manuscript is interesting. However, there is lack of information in the manuscript and the authors need to clarify following question and concern.
For the destruction of TezRs, authors treated various cells with nucleases. But They have not directly demonstrated whether the nuclease completely destroys TezRs or only extracellular TezRs. The controls in this experiment were not treated with nuclease and are not directly comparable to those with or without TezRs. It should be shown that TezRs are lost, by treatment, in each cell type used in this study. It would also be necessary to have a control to show that other functions of that cell were not lost by the treatment.
Author Response
Question: For the destruction of TezRs, authors treated various cells with nucleases. But They have not directly demonstrated whether the nuclease completely destroys TezRs or only extracellular TezRs.
Response: We thank the reviewer for this comment and we apologize that this part was unclear. In our study we found that there are two types of extracellular TezRs. Primary TezRs are extracellular, cell-surface bound nucleic acids which can’t be removed from the cells following centrifugation and the removal of medium or matrix. Secondary TezRs are also extracellular, but can be easily washed out along with culture medium or matrix. Trying to address the reasons for the unique behavior of cells following the loss of both DNA- and RNA-based primary TezRs, we speculated it could lead to the activation of some proposed internal (i.e. cytoplasmic) TezRs (TezR–i). At the same time, we have not identified the presence of TezR-i per se. However, based on our proprietary results and based on the literature data, nucleases used in this study did not penetrate the cells. Therefore, by using treatment regimens of nucleases, we could destroy only extracellular TezRs and not internal TezRs (TezR–i).
In terms of efficacy the efficacy of the used regimens of nuclease treatment (new Supplementary Figure 1A,B) we found that cells treated with either DNase or RNase alone exhibited a decrease, but not the total disappearance of fluorescence compared to control cells. However, cells treated with a combination of DNase and RNase revealed the total disappearance of surface fluorescence compared to single-nuclease treatment. Therefore, we believe that with these fluorescence-based studies we were able to show that all cell-surface bound DNA and RNA molecules were removed from the cell. Moreover, the alteration of cell responses following the treatment of cells with nuclease is the primary line of evidence that nucleases treatment regimens are sufficient to alter the regulatory role of TezRs. In other words, the treatment was enough to modulate cell response regardless of whether there were any remaining TezRs.
Question: The controls in this experiment were not treated with nuclease and are not directly comparable to those with or without TezRs.
Response: Indeed, in our study “control” cells were not treated with nucleases, but were treated with water (that was used as a vehicle for nucleases. We are sorry since due to our negligence in the several parts of manuscript we used the term “untreated cells”, meaning that they were not treated with nucleases; however, these cells were treated with water and were processed the same way as the experimental probes PBS. So we made the necessary corrections in the manuscript.
Moreover, in all studies when we analyzed cells response following the loss of cell-surface bound (primary) TezRs, cells were isolated from the extracellular matrix (to remove secondary TezRs) and were treated with nucleases. Control cells were processed the same way, under the same conditions but were treated with water. Such an approach allowed us to compare the response of cells following the loss of the primary TezRs under the same growth conditions as the control cells.
We are sorry that this part was not clear and added a clarification to the Materials and Methods section:
Lines 125-127 “Control cells were processed in the same manner; however, instead of treatment with nucleases, they were treated with water (used as a vehicle for nucleases).”
Lines 139-140 “Control C. albicans were cultivated on the agar without nucleases.”
Line 148 “Controls were similarly treated with water (used as a vehicle for nucleases)..”
Line 275-277 “Control probes were treated with water used as a vehicle for nucleases , and cells were placed in 2 ml microcentrifuge tubes”
Line 291-293 “C. albicans treated with water (used as a vehicle for nucleases) was used as a control and was processed in the same way and heated at the lowest non-tolerable temperature.”
Line 300-301 “The control probes were treated with water”
Results:
Lines 508 “All cells were treated with water (control)…”
Lines 518-519 “S-phase after the loss of primary DNA-formed TezRs compared with water-treated control.”
Lines 562-563 “Data represent the averages of three independent experiments and were compared with the water-treated control.”
Lines 571-573 “When migration rate kinetics in the early stages of wound closure were studied, it was observed that cells with destroyed TezRs started moving into the gap much earlier than control, water-treated cells (Figure 3B).”
Line 580-583 “The fastest kinetics were observed for Vero TezR–D1d/R1d cells with 43%, 56%, and 72% wound closure at 8, 12, and 24 h of growth, respectively, compared with those in water- treated cells, which displayed 20%, 45%, and 48% closure, respectively, at the same time points (p<0.05,’
Lines 592-593 “The effects of primary TezR loss or inactivation with PI was compared with water-treated control cells. ”.
Line 678-679 “The inactivation of any primary TezRs resulted in growth retardation with an extended time lag compared with that in vehicle-treated C. albicans….”
Lines 701-704 “The percentages of the different morphological forms were quantified using the morphological index (MI) of individual cells after 72 h of growth at 37ºC compared with water treated control (Figure 6A, B).”
Line 705-706 “The MI for water treated cells revealed mixed yeast, pseudohyphal, and hyphal morphologies”
Lines 817-818 “Control Vero cells were infected with HSV-1 at an MOI of 0.1. They were either treated with water (control), or primary TezRs were destroyed 2 h post-infection (h.p.i.).”
Lines 828-829 “Morphological analysis revealed that treatment of water -treated control cells with unaltered TezRs with ITS in FDS-free medium….”
Lines 881-882 “We observed phototoxic effects on water -treated Vero cells with only a few cells adhering after growth in the light environment (Fig 9A,B).”
Line 883-884 “Vero TezR–D1d cells exhibited the same pattern of phototoxic effects in a light environment as that of vehicle-treated cells.”
Lines 907 “Water-treated, control C. albicans exhibited maximum tolerance up to 55°….”
Lines 912-914 “By analyzing the dynamics of fungal cell death at different temperatures, we found that control, water-treated C. albicans had a reduced CFU number by more than 4 log10 units when….”
Lines 990-991 increased growth of control, water-treated C. albicans with an almost 4-fold increase in the OD600 value.”
Lines 1152-1153 “from TezR–D1d seeds, which represented an improvement of approximately 57.9% and 27.5%, as compared to those for water control samples (p<0.05).”
Lines 1239 “based TezR, as well other individual patterns or even those of control cells.”
Question: It should be shown that TezRs are lost, by treatment, in each cell type used in this study. It would also be necessary to have a control to show that other functions of that cell were not lost by the treatment.
With respect to the visualization that TezRs were lost following the treatment with nucleases, since nucleases under the used conditions do not penetrate inside the cells their effects could only be realized extracellularly. We added the representative images (supplementary figure 1A, B) of the alteration of the fluorescence of CHO and Vero cells following the treatment with used nucleases. We would like to highlight that the manuscript included a lot of data, and we tried to keep some parts shorter. Therefore, given that we observed the disappearance of cell-surface bound nucleic acids with the selected regimens of nucleases resulted in the alteration of characteristics of all cell cultures used in the study (Vero, CHO, HEK293, T98G), we believe that visualization of the loss of cell-surface bound nucleic acids with nucleases on Vero and CHO cells (new Supplementary Figure 1A, B) would be sufficient, and that the used regimens of nucleases will demonstrate a similar effect the of the loss of cell-surface bound nucleic acids on other cell lines as well.
Lines 483-494 “We confirmed the presence of cell surface-bound nucleic acid based TezRs through the analysis of altered fluorescence in Vero or CHO cells following the removal of the ex-tracellular matrix and their treatment with treated with water (control) or a combination of DNase I and RNase at a concentration of 10 µg/mL for 10 minutes. Control PI stained cells displayed clear fluorescence around the cells, and those stained with Sytox 9 re-vealed fluorescence on the cell surface as well as nucleus, confirming the presence of cell surface-bound nucleic acids, which remained after the removal of culture medium (Supplementary Fig. 1 A, B).
For both cell lines, cells treated with either DNase or RNase alone exhibited a decrease in fluorescence around the cells compared to vehicle-treated cells. However, cells treated with a combination of DNase and RNase displayed the total disappearance of surface fluorescence.”
Regarding the control that “other functions of cell are not altered following the treatment with nucleases.” It is important to mention that cells following the loss of DNA-based, RNA-based or both DNA and RNA-based TezRs had altered responses to various environmental factors (also clearly seen in transcriptome analysis). And we used altered chemical or physical conditions to visualize this altered response. So in our study, we observed that multiple functions of the cells were altered following the loss of particular TezRs.
Reviewer 2 Report
Comment:
This paper discusses "Novel cell receptor system of eukaryotes formed by previously 2 unknown nucleic acid-based receptors ". The main contribution of the paper is "the first evidence for the existence of a previously unknown receptive system formed by novel DNA- and RNA-based receptors in eukaryotes. This system, named the TR-system, is capable of recognizing and generating a response to different environmental factors and has been shown to orchestrate major vital functions of fungi, mammalian cells, and plants"
This is an interesting study and is generally well written and structured.
Minor comments:
· Well written except in some situations. I advise recheck it again.
· The introduction should be advised to be re-written to be in more logical flow.
· The methods in details should be described and analysis as well.
· Please, Suggest future experiments in details
· Please, try to add general paragraph about importance of GPCRS.
· Generally , Figures 3 and 4, 14, 15 are not obvious to me. It is not understandable.
· Although it needs to be in more logical flow, the introduction provides a good, generalized background of the topic. However, why not cite more literature papers .
· I think the motivations for this study need to be made clearer
· Regarding the figures: I recommend make more figures to be illustrative.
· PCA is not obvious. Explain more
Given these shortcomings the manuscript requires Minor revisions.
"I recommend that this paper be accepted after minor revision."
Author Response
Minor comments:
- Question: Well written except in some situations. I advise recheck it again.
Response: We thank the reviewer for this comment. We reread the manuscript and made the following modifications to improve readability.
Lines 532-533 “Next, we examined the role of TezRs in mammalian cell growth and plating using Vero cells whose viability was not affected by the loss of primary TezRs.”
Lines 538-541 “The destruction of TezR–R1 resulted in significantly decreased cell sizes (p<0.001) in cells after 48 h, whereas the individual destruction of TezR–D1 or combined destruction of TezR–D1/R1 had no significant effect on cell size measurements (Figure 2B).”
Lines 571-573 “When migration rate kinetics in the early stages of wound closure were studied, it was observed that cells with destroyed TezRs started moving into the gap much earlier than control, water-treated cells (Figure 3B).
Lines 611-620 “To address the differences in transcriptome profile between Vero cells following the loss of any primary TezRs and control cells, principal component analysis (PCA) was first used to study potential clusters based on differently detected genes. The PCA plot visually showed that PC1 and 2 separated the Vero control, TezR-D1d, TezR-Rd and TezR-D1d/R1d as four distinctive clusters (Fig. 4A). Although the largest difference in PCA was observed in cells with destroyed TezR–R1d , this PCA result indicated that Vero cells following the loss of any primary TezRs could be separated by their transcriptome profile by clustering. ”
Lines 621-626 “Next, we compared the results from each probe and analyzed the genes for which expression was significantly altered (upregulated or downregulated) following the removal of different TezRs. Volcano plots of the log2(fold change) of DEGs (|log2 fold-change| > 0.5 and p-value < 0.05) show that many of the differentially upregulated and downregulated changes were highly significant in Vero cells lacking TezRs compared to control cells (Fig. 4B to D, Supplementary table S1).”
Lines 1009-1012 “We compared the lag phase, which was taken as the period between inoculation of fungi and the initiation of biomass growth, and included the time required for sensing maltose and gene expression changes required to start its utilization (71, 72).”
Lines 1098-1101 “In this experiment, we used low doses of tenofovir, lamivudine, and etravirine, which were at least a 100-fold less than their MICs and did not show any inhibitory effects against C. albicans (the MICs of these RTIs were over 500 µg/ml as the highest concentration tested) (Supplementary Table 2). ”
Lines 1106-1107 “Studying the effect of RTIs on C. albicans with inactivated TezRs, we found that they had a statistically significant effect on cell growth with destroyed TezR–R1 (Figure 14A).”
Lines 1123-1127 “Thereafter, we studied how different periods of raltegravir addition from 0 to 120 minutes could affect maltose utilization. We found that raltegravir inhibited C. albicans growth on M9 with maltose when added together or within the first time periods but stopped having an inhibiting role once it was added starting from 120 min of growth (Figure 14B).”
1164-1169 “Effect of TezR inactivation in seeds of Triticum aestivum on cultivation. (A) Morphological appearance and growth differences of Triticum aestivum seedlings after DNA, RNA- or DNA- and RNA-based TezR inactivation, measured on day 7. The bar on the right shows the length in cm. (B) Average stem and root length (cm) on day 7 as the mean±SD. (C) Chlorophyll a, b, and total chlorophyll (mg/g) levels on day 7.”
- Question: The introduction should be advised to be re-written to be in more logical flow.
Response: The authors appreciate the reviewer’s comment. The introduction has been revised to make it more easy to follow.
- Question: The methods in details should be described and analysis as well.
Response: The authors appreciate the reviewer’s comments. The relevant sections were rewritten and more methodological details were added throughout the Materials and Methods section
- Question: Please, Suggest future experiments in details
Response: The list of future experiments was clarified and expanded to include (lines 1393-1403) “Future studies should unveil in more detail, the molecular mechanisms underlying the origin and the pathway for the translocation of TezRs to the cell surface, their specific effects on different cells including immune and malignant cells in addition to expanding our understanding of the entire set of sensing and regulatory processes in which they are involved. The interaction between TezRs with known protein-based receptors performing the same function is also a critical question that might pave the way to finding a more accurate way to control cell signaling and reception. Given that primary TezRs are associated with the cell membrane, they might also be involved in the functioning of ion channels; meaning that they can be used to better regulate cell membrane potential and play a role in neural communication and muscle contraction. Finally, the role of TezRs in the response of multicellular organisms is also intriguing.”
Question: Please, try to add general paragraph about importance of GPCRS.
Response: Per the reviewer’s suggestion we added lines 37-40 with relevant references
- Question: Generally, Figures 3 and 4, 14, 15 are not obvious to me. It is not understandable.
Response: We are sorry that the representation of these images was not clear due to some negligence on our side. We added additional clarifications to make these figures more clear.
Specifically:
Figure 3.
Following the Reviewers’ recommendation, we found that we failed to include a relevant description of Figure 3b. So, we added lines 598-601 “Heatmap showing the differences in migration rates. The color intensity of percentage wound closure, is represented by a color scale, from white (minimal) to red (maximum). Data represent the average of three independent experiments.” Moreover, we found that we had not included the legend to Figure 3B, which is now added.
Figure 4.
To better describe PCA, the following parts were rewritten in the text
Lines 611-620 “To address the differences in transcriptome profile between Vero cells following the loss of any primary TezRs and control cells, principal component analysis (PCA) was first used to study potential clusters based on differently detected genes. The PCA plot visually showed that PC1 and 2 separated the Vero control, TezR-D1d, TezR-Rd and TezR-D1d/R1d as four distinctive clusters (Fig. 4A). Although the largest difference in PCA was observed in cells with destroyed TezR–R1d , this PCA result indicated that Vero cells following the loss of any primary TezRs could be separated by their transcriptome profile by clustering.”
Lines 623-626 “Volcano plots of the log2(fold change) of DEGs (|log2 fold-change| > 0.5 and p-value < 0.05) show that many of the differentially upregulated and downregulated changes were highly significant in Vero cells lacking TezRs compared to control cells (Fig. 4B to D, Supplementary table S1).
Figure 14.
Response: We thank the reviewer for this comment. To figure 14A we have added a Y axis label “hours” and “log10 (CFU/mL)” label to the legend.
We also revised figure 14B including the text to make it more clear (lines 1123-1127) “Thereafter, we studied how different periods of raltegravir addition from 0 to 120 minutes could affect maltose utilization. We found that raltegravir inhibited C. albicans growth on M9 with maltose when added together or within the first time periods but stopped having an inhibiting role once it was added starting from 120 min of growth (Figure 14B).”
Figure 15.
We added additional description line 1165-1169 “Effect of TezR inactivation in seeds of Triticum aestivum on cultivation. (A) Morphological appearance and growth differences of Triticum aestivum seedlings after DNA, RNA- or DNA- and RNA-based TezR inactivation, measured on day 7. The bar on the right shows the length in cm. (B) Average stem and root length (cm) on day 7 as the mean±SD. (C) Chlorophyll a, b, and total chlorophyll (mg/g) levels on day 7..” We also found the misprints in the axis names in images 15B and C, that have now been corrected.
- Question: Although it needs to be in more logical flow, the introduction provides a good, generalized background of the topic. However, why not cite more literature papers.
Response: As suggested additional references were added:
Hakak, Y. et al, Global analysis of G‐protein‐coupled receptor signaling in human tissues. FEBS letters. 550, 11-17, 2003.
Salon, J.A., Lodowski, D.T., & Palczewski, K. The significance of G protein-coupled receptor crystallography for drug discovery. Pharmacol rev. 63, 901-937 (2011).
Basith, S. et al. Exploring G protein-coupled receptors (GPCRs) ligand space via cheminformatics approaches: impact on rational drug design. Front pharmacol 9, 128 (2018). 21.25. Ikeya, N., & Woodward, J.R. Cellular autofluorescence is magnetic field sensitive. PNAS. 118, e2018043118. (2021).
Klinkert, B., & Narberhaus, F. Microbial thermosensors. Cell. Mol. Life Sci. 66, 2661-2676 (2009).
Ahmed, R., & Gray, D. Immunological memory and protective immunity: understanding their relation. Science. 272, 54-60 (1996).
Wolf DM, Fontaine-Bodin L, Bischofs I, Price G, Keasling J, Arkin AP. Memory in microbes: quantifying history-dependent behavior in a bacterium. PLOS one. 3, e1700 (2008).
Lambert, G., & Kussell, E. Memory and fitness optimization of bacteria under fluctuating environments. PLoS genetics. 10, e1004556 (2014).
Tetz, V., & Tetz, G. Bacterial DNA induces the formation of heat-resistant disease-associated proteins in human plasma. Sci Rep. 9, 1-0 (2019).
Tetz, G., Pinho, M., Pritzkow, S., Mendez, N., Soto, C., & Tetz V. Bacterial DNA promotes Tau aggregation. Sci Rep. 10, 1-1 (2020).
Tetz, G., & Tetz, V. Bacterial extracellular DNA promotes β-amyloid aggregation. Microorganisms 9, 1301 (2021)
115. Gyula, P. et al. Ambient temperature regulates the expression of a small set of sRNAs influencing plant development through NF‐YA2 and YUC2. Plant Cell Environ. 10, 2404-2417 (2018).
Gershman, S.J., Balbi, P.E., Gallistel, C.R., and Gunawardena, J. Reconsidering the evidence for learning in single cells. Elife. 10:e61907 (2021).
Cools-Lartigue J et al. Neutrophil extracellular traps sequester circulating tumor cells and promote metastasis. J. Clin. Invest. 123, 3446-58 (2013).
- Question: I think the motivations for this study need to be made clearer
Response: This section was clarified and the sentence “Taking a broad range of stimuli whose reception can’t be well explained with known receptors we studied the receptive and regulatory roles of extracellular DNA and RNA, since the novel roles, of nucleic acids that go beyond the processing of genetic information and protein synthesis, including the interaction with proteins, were recently described (48-50). We discovered that prokaryotes have a previously unknown receptive system that is responsible for the interaction of cells with the environment. Here, we show the existence of a similar, previously unknown TR-receptive system in eukaryotes that manages interactions between cells and the environment.” was added in (lines 72-79) to make motivations for this study clearer.
- Question: Regarding the figures: I recommend make more figures to be illustrative.
Response: We understand the reviewer’s comment; however, there are certain limitations to the total number of figures allowed in the manuscript per the journal’s guidelines. As per Reviewer’s comments we added two supplementary images (Supplementary figure 1A and 1B).
- Question: PCA is not obvious. Explain more
Response: We are sorry for being unclear. This section was clarified and the sentence “To address the differences in transcriptome profile between Vero cells following the loss of any primary TezRs and control cells, principal component analysis (PCA) was first used to study potential clusters based on differently detected genes. The PCA plot visually showed that PC1 and 2 separated the Vero control, TezR-D1d, TezR-Rd and TezR-D1d/R1d as four distinctive clusters (Fig. 4A). Although the largest difference in PCA was observed in cells with destroyed TezR–R1d , this PCA result indicated that Vero cells following the loss of any primary TezRs could be separated by their transcriptome profile by clustering..” was added in (lines 611-620) to explain PCA in more detail.
Round 2
Reviewer 1 Report
My comments and questions have been adequately answered and the paper has been appropriately revised.